# GFT: Graph Foundation Model with Transferable Tree Vocabulary

**Zehong Wang**
University of Notre Dame
Indiana, USA
zwang43@nd.edu

**Zheyuan Zhang**
University of Notre Dame
Indiana, USA
zzhang42@nd.edu

**Nitesh V Chawla**
University of Notre Dame
Indiana, USA
nchawla@nd.edu

**Chuxu Zhang**\*
University of Connecticut
Connecticut, USA
chuxu.zhang@uconn.edu

**Yanfang Ye**\*
University of Notre Dame
Indiana, USA
yye7@nd.edu

## Abstract

Inspired by the success of foundation models in applications such as ChatGPT, as graph data has been ubiquitous, one can envision the far-reaching impacts that can be brought by Graph Foundation Models (GFMs) with broader applications in the areas such as scientific research, social network analysis, drug discovery, and e-commerce. Despite the significant progress of pre-trained graph neural networks, there haven't been GFMs that can achieve desired performance on various graph-learning-related tasks. Building GFMs may rely on a vocabulary that encodes transferable patterns shared among different tasks and domains. Unlike image and text, defining such transferable patterns for graphs remains an open question. In this paper, we aim to bridge this gap by rethinking the transferable patterns on graphs as computation trees – i.e., tree structures derived from the message-passing process. Based on this insight, we propose a cross-task, cross-domain graph foundation model named GFT, short for Graph Foundation model with transferable Tree vocabulary. By treating computation trees as tokens within the transferable vocabulary, GFT improves model generalization and reduces the risk of negative transfer. The theoretical analyses and extensive experimental studies have demonstrated the transferability of computation trees and shown the effectiveness of GFT across diverse tasks and domains in graph learning. The open source code and data are available at https://github.com/Zehong-Wang/GFT.

## 1 Introduction

Foundation models such as Large Language Models (LLMs) and Large Vision Models (LVMs) keep reshaping our view of the world [7, 100, 51, 112, 50]. These models are designed to be general-purpose, adaptable across various tasks and domains through fine-tuning or prompting, such as GPT-4 [1] in Natural Language Processing (NLP) and Sora [46] in Computer Vision (CV). Research attributes the success of foundation models to the uniformity of tasks and a general vocabulary that defines basic transferable patterns across tasks [98, 76, 112, 3, 50]. For example, LLMs [1, 112] treat language tasks as question-answering or next-word prediction and deconstruct sentences using a word vocabulary. Similarly, LVMs [100, 98, 3] reformulate image tasks as image question-answering, converting images into discrete tokens using a vision vocabulary. Inspired by the success of LLMs

---

\*Corresponding Authors.

38th Conference on Neural Information Processing Systems (NeurIPS 2024).

and LVMs, as graph-structured data (e.g., citation networks, social networks, computer networks, molecular structures, and recommender systems) have become ubiquitous, one can envision the far-reaching real-world impacts that can be brought by pre-trained Graph Foundation Models (GFMs).

Although there has been significant progress of pre-trained Graph Neural Networks (GNNs), there haven't been GFMs that can achieve desired performance on a wide range of graph-learning-related tasks. Unlike CV and NLP, as graphs represent complex, non-Euclidean relationships among entities [92, 48, 104, 58, 107], a grand challenge of building GFMs lies in identifying transferable patterns across graphs [50, 93, 25]. There have been extensive studies aiming to tackle this challenges, which can mainly be categorized into two groups: (1) Utilizing graphon theory: Ruiz et al. [62] provide theoretical evidence of transferability between two graphs generated from the same graphon. Cao et al. [8] further extend these findings by both empirically and theoretically analyzing graph transferability in pre-training and fine-tuning scenarios. Despite these theoretical guarantees, the stringent assumptions of graphon theory often prove difficult to satisfy in real-world, cross-domain datasets [42], thus limiting its applicability in defining transferable graph vocabularies. (2) Exploring graph transferability using subgraph structures [114, 59, 50]: Zhu et al. [114] demonstrate that the transferability between graphs is linked to the ego-graph patterns of individual nodes and establish a stability bound using the graph Laplacian. They suggest that localized subgraphs could serve as transferable patterns within graph vocabularies. Building on this finding, Sun et al. [68] develop a GFM by reformulating graph tasks as subgraph classification, enabling a single model to be applied to diverse tasks. Huang et al. [30], Liu et al. [45] expand GFMs to cross-domain scenarios by unifying the node feature space across different graphs through LLMs [60, 76]. Despite these successes, the process of explicit subgraph extraction remains time and memory intensive [30]. More importantly, numerous studies such as [20, 10, 53, 103] show that message-passing GNNs [40, 24, 21] fail to detect critical substructures or motifs within subgraphs, reducing the feasibility of using subgraphs to define graph vocabularies.

***How to identify a vocabulary that can encode transferable patterns shared among different tasks and domains for the construction of GFMs?*** In this paper, we aim to address the limitations of existing works by answering this question. Specifically, based on message-passing mechanism of GNNs, we have observed that the learned embeddings of each node can be essentially captured in the form of its computation tree. Based on this insight, we bridge the research gap by rethinking the transferable patterns on graphs as computation trees – i.e., subtree structures derived from the message-passing process. Using computation tree as a transferable pattern across graphs will bring three primary advantages: (1) *Efficiency*: As the extraction and encoding of computation trees are integrated within a single message-passing GNN process [20], it eliminates the need for the explicit subgraph extraction for GFMs [30, 45]. (2) *Expressiveness*: Since computation trees are capable of capturing localized patterns [52], it's able to represent a graph as a multiset of computation trees [23]. (3) *Learnability*: As the information of computation trees is completely captured by message-passing GNNs, it can tackle the issue that certain motifs within subgraphs remain elusive. We theoretically investigate the transferability of computation trees and empirically demonstrate a strong correlation between computation tree similarity and transfer learning performance across various graphs.

Based on the key idea above, by treating computation trees as graph vocabulary tokens, we develop a cross-task, cross-domain graph foundation model – namely GFT – short for Graph Foundation model with transferable Tree vocabulary. GFT consists of pre-training and fine-tuning phases, enabling it to handle datasets across different tasks and domains effectively. During pre-training, we introduce a computation tree reconstruction task to acquire generalized knowledge from cross-domain graphs. We obtain a discrete tree vocabulary of prototypical tree tokens by quantizing the embedding space of computation trees, which theoretically improves model generalization. In the fine-tuning phase, we utilize this learned tree vocabulary to unify various graph-related tasks into computation tree classification, thereby preventing negative transfer [89, 87]. Extensive experimental results demonstrate the effectiveness of GFT in graph learning on cross-task and cross-domain datasets.

## 2 Rethinking Transferable Patterns on Graphs

### 2.1 Transferability on GNNs

Transferability refers to a model's capability to extract patterns from source tasks and apply this knowledge to enhance performance on related target tasks [5, 33, 90]. Understanding transferable

patterns is essential for developing graph foundation models. Early research focuses on analyzing transferability through the perspectives of graph spectrum [41, 42] and subgraphs/substructures [114], defining transferability as model invariance to minor permutations on the graph. A more recent study [50] investigates the transferable vocabulary in graphs by identifying key substructures relevant to various tasks. For instance, they find that triadic closure, homophily, and heterophily are vital for node classification; local and global structural proximities are crucial for link prediction; and certain motifs [103], such as triangles, $k$-cliques, and stars, serve as fundamental components for graph classification. Another line of research [62, 8, 69] incorporates graphon theory to provide a theoretical basis for transferability. Specifically, Ruiz et al. [62] establish a bound on the embeddings of two graphs sampled from the same graphon. Cao et al. [8] expand this to include pre-training and fine-tuning scenarios, assessing the distance between graphs based on their alignment within the graphon space. However, the stringent assumptions of graphon theory limit its practical application in the design of graph foundation models.

We identify two primary limitations in analyzing transferable patterns on graphs: (1) While certain domains [110, 87, 43, 108] or tasks [114, 111, 19] exhibit transferable patterns, the challenge of identifying universally transferable substructures is difficult. (2) More critically, basic message-passing GNNs, constrained by the 1-WL test [94, 52], fail to recognize certain subgraphs (or motifs) [20, 10, 103], such as stars, conjoint cycles, and $k$-cliques, as well as heterophily patterns [113]. This limitation in recognizing substructures impedes using subgraphs as transferable tokens in graph vocabulary [68, 30, 45]. More related works are elaborated in Appendix A.

## 2.2 Computation Tree as Transferable Pattern

In this paper, we rethink the transferable pattern in graphs as the computation tree — a specialized subtree pattern that emerges from unfolding the message-passing process [12]. This pattern is demonstrably effective at capturing critical localized information within the graph [20, 52, 94, 12]. Treating computation trees as tokens within a graph vocabulary offers two distinct advantages: (1) computation trees preserve the essential structural information of the graph, which is learnable through message-passing GNNs, and (2) the computation tree structure occurs across various graph-based tasks. These tasks can be unified as computation tree classification by integrating a virtual node, as shown in Figure 1.

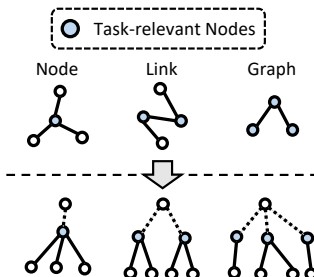

Figure 1: Graph tasks (top) and the corresponding computation trees (bottom). A virtual node can be added at the top to connect all task-relevant nodes, unifying different tasks as the tree-level task.

Before diving into transferability analysis, we first establish the necessary notations. Consider a graph $\mathcal{G} = (\mathcal{V}, \mathcal{E})$ composed of node set $\mathcal{V}$ and edge set $\mathcal{E}$. Each node $v \in \mathcal{V}$ is associated with a feature vector $\mathbf{x}_v \in \mathbb{R}^d$ and a computation tree $\mathcal{T}_v$ with $L$ layers. A GNN encoder $\phi$ processes these computation trees as inputs, producing embeddings for root nodes $\mathbf{z} = \phi(\mathcal{T}_v) \in \mathbb{R}^{d'}$.

**Definition 2.1** (Computation Trees [12]). Given a graph $\mathcal{G} = (\mathcal{V}, \mathcal{E})$, define $\mathcal{T}_v^1 = v$ and $\mathcal{T}_v^L$ as the $L$-layer computation tree. This tree is constructed by recursively integrating the subtrees of neighborhoods. The multiset of $L$-layer computation trees on graph $\mathcal{G}$ is denoted by $\mathcal{T}_{\mathcal{G}}^L := \{\mathcal{T}_v^L\}_{v \in \mathcal{V}}$.

Figure 1 demonstrates the construction of computation trees across various graph tasks, including node-, link-, and graph-level tasks. These trees capture essential localized subtree patterns within the graphs [55, 64, 12]. If the $L$-layer computation trees for two nodes are similar, it indicates that these nodes share similar neighborhoods, suggesting they represent analogous phenomena [42]. Thus, it is rational to assess transferability of computation trees by examining the stability of GNNs in producing analogous embeddings for similar trees [62, 42].

**Theorem 2.2** (Transferability of Computation Tree ). *Given two $L$-layer computation trees $\mathcal{T}_{v_1}, \mathcal{T}_{v_2}$ derived from the graph $\mathcal{G}$ and a GNN encoder $\phi$, the Euclidean distance between the tree embeddings $\Delta \triangleq \|\phi(\mathcal{T}_{v_1}) - \phi(\mathcal{T}_{v_2})\|_2$ is bounded as follows:*

$$\Delta \leq \mathcal{C}_1 \|\mathbf{x}_{v_1} - \mathbf{x}_{v_2}\|_2 + \mathcal{C}_2 \sum_{j \in \mathcal{N}(v)} \Delta_{v_1, v_2, j}^{L-1} \leq 2\mathcal{B}_{\mathbf{x}}(\mathcal{C}_1 + \sum_{l=1}^{L} \mathcal{C}_2^l D_l) \tag{1}$$

where $\Delta_{v_1,v_2,j}^{L-1}$ represents the distance between the $(L-1)$-layer subtrees of the $j$-th children of nodes $v_1$ and $v_2$, $\mathcal{C}_1, \mathcal{C}_2$ are constants, and $\mathcal{B}_{\mathbf{x}}$ denote bounded norm of $\mathbf{x}$. The variable $d_l$ indicates the number of children in the $l$-layer subtrees, and $D_l = d_l d_{l-1}...d_1$.

*Proof.* All proofs in the paper are detailed in Appendix D. $\square$

*Remark* 2.3. Theorem 2.2 derives a recursive bound for computation tree similarity. In particular, the distance between two computation trees is closely correlated to the similarity of their subtrees, where higher subtree similarity results in a closer distance. This suggests that computation trees with similar structures are likely to have similar embeddings, which enhances their transferability [114, 33, 62]. This aligns with our empirical observations that higher computation tree similarity between two graphs leads to improved transferability.

**Supportive Observations — Synthetic Graphs.** Figure 3 shows that high computation tree similarity between graphs correlates with improved transfer learning performance on synthetic graphs (Figure 2). Specifically, we construct three distinct graphs: $\mathcal{G}_1$ and $\mathcal{G}_2$ share similar motifs but differ in computation tree distributions, while $\mathcal{G}_1$ and $\mathcal{G}_3$ exhibit dissimilar motifs but similar computation tree distributions. We employ the WL subtree kernel [64] and the graphlet sampling kernel [57] to assess tree and motif similarity, respectively, and utilize the inverse of the Central Moment Discrepancy [102] to measure transferability. Further details on experimental settings and additional results are available in Appendix E.1. Our findings indicate that transferability is strongly associated with computation tree similarity rather than motif similarity, regardless of the scale of graphs (# blocks).

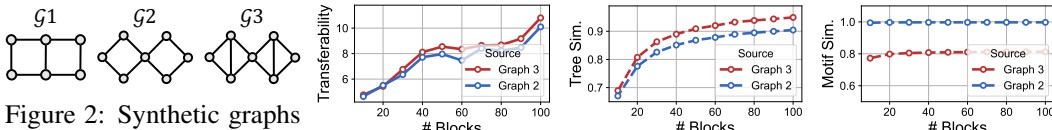

Figure 2: Synthetic graphs composed of two basic blocks. More blocks can scale up the graph sizes.

Figure 3: Transfer performance on synthetic graphs with $\mathcal{G}_1$ as the target graph. Higher tree similarity correlates with enhanced transferability.

**Supportive Observations — Real-world Graphs.** Table 1 validates the correlation between computation tree similarity and transferability in real-world graphs, including homophily `Airport` networks [61] and heterophily `WebKB` networks [56]. We evaluate transferability based on transfer learning performance in node classification tasks. Detailed experimental settings and additional results are available in Appendix E.2. Our findings in real-world graphs corroborate those in synthetic graphs: higher computation tree similarity enhances transferability, while the impact of motifs is marginal, no matter using original node features (Table 1) or randomized node features (Table 9).

Table 1: Transfer learning performance on homophily (above) and heterophily (below) graphs. For any target graph, source graphs with higher tree similarity lead to improved accuracy, highlighted with Blue . Conversely, the influence of motif similarity is marginal, marked by LightBlue .

| $\mathcal{G}_{target} \rightarrow$ | Brazil | | Europe | | USA | |
|---|---|---|---|---|---|---|
| $\mathcal{G}_{source} \rightarrow$ | Europe | USA | Brazil | USA | Brazil | Europe |
| Motif Sim. | 99.01 | 92.65 | 99.00 | 96.81 | 92.68 | 96.81 |
| Acc. / Tree Sim. | 53.1 / 34.6 | 56.8 / 62.2 | 50.8 / 34.6 | 51.4 / 88.7 | 54.5 / 62.2 | 57.9 / 88.7 |

| $\mathcal{G}_{target} \rightarrow$ | Cornell | | Texas | | Wisconsin | |
|---|---|---|---|---|---|---|
| $\mathcal{G}_{source} \rightarrow$ | Texas | Wisconsin | Cornell | Wisconsin | Cornell | Texas |
| Motif Sim. | 99.97 | 99.98 | 99.99 | 99.99 | 99.98 | 99.99 |
| Acc. / Tree Sim. | 46.5 / 65.3 | 42.4 / 42.7 | 56.0 / 65.3 | 53.1 / 41.7 | 48.6 / 42.7 | 48.2 / 41.7 |

## 3 GFT: Graph Foundation Model with Transferable Tree Vocabulary

We develop GFT, a cross-domain and cross-task graph foundation model that leverages computation trees as transferable patterns within graph vocabulary. As illustrated in Figure 4, GFT undergoes

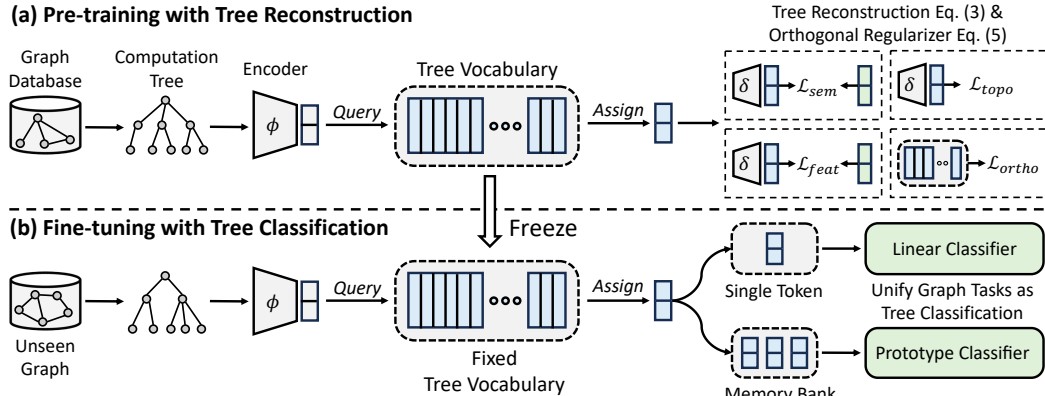

Figure 4: During pre-training, GFT encodes general knowledge from a graph database into a tree vocabulary through tree reconstruction. In fine-tuning, the learned tree vocabulary is applied to unify graph-related tasks as tree classification, adapting the general knowledge to specific tasks.

pre-training through a computation tree reconstruction task to acquire general knowledge from a cross-domain graph database. Subsequently, GFT quantizes the embedding space of computation trees to form a discrete tree vocabulary, encapsulating fundamental, transferable computation tree patterns for diverse tasks. In the fine-tuning phase, GFT utilizes this tree vocabulary to unify graph-related tasks (including node-, link-, and graph-level tasks) as computation tree classification, adapting the general knowledge to specific target tasks.

## 3.1 Pre-training with Computation Tree Reconstruction

The pre-training stage focuses on learning general computation tree patterns on graphs, facing two primary challenges: (i) obtaining transferable patterns, and (ii) comprehensively capturing computation tree knowledge. For the first challenge, we learn a discrete tree vocabulary by quantizing the embedding space of computation trees [77]. For the second challenge, we introduce a computation tree reconstruction task that considers multiple aspects of computation trees.

**Learning Tree Vocabulary.** The idea of learning a discrete computation tree vocabulary originates from the principles of sparse distributed memory in cognitive science [37, 38], which stores and retrieves memory in a distributed manner. By adopting these principles, the tree vocabulary maintains a set of tokens that are reusable and adaptable across various tasks, improving model transferability.

We adopt the Vector Quantization (VQ) [77] to develop the tree vocabulary. Given a graph database[2] $\mathcal{D} = \{\mathcal{G}_i\}_{i=1}^n$, we randomly extract a set of computation trees $\mathcal{T} = \{\mathcal{T}_i\}_{i=1}^m$ and employ a GNN encoder $\phi$ to generate the tree embeddings $\mathcal{Z} = \{\mathbf{z}_i\}_{i=1}^m$. We define the computation tree vocabulary as a set of learnable tokens $\mathbf{C} = \{\mathbf{c}_1, ..., \mathbf{c}_K\}$. The tree embedding space is quantized by assigning each embedding to the nearest token, resulting in quantized tree embeddings $\mathbf{q}_i = \mathbf{c}_j$, where $j = \arg\min_j \|\mathbf{z}_i - \mathbf{c}_j\|_2$. We optimize this projection by back-propagating the reconstruction error to the tree vocabulary $\mathbf{C}$ and applying a straight-through gradient estimator [6] to the encoder $\phi$. In particular, we jointly optimize vocabulary loss and commitment loss [77], along with tree reconstruction loss (discussed later), where the former updates the token vectors $\mathbf{c}$ using the fixed quantization $\mathbf{q}$, and the latter ensures alignment between the tokens in the vocabulary and the quantized tree embeddings, serving as a regularizer. The pre-training objective is thus defined as:

$$\mathcal{L}_{pretrain} = \mathcal{L}_{tree} + \underbrace{\frac{1}{m}\sum_{i=1}^m \left\|\mathrm{sg}[\mathbf{z}_i] - \mathbf{c}_i\right\|_2^2}_{\text{vocabulary loss}} + \beta_1 \cdot \underbrace{\frac{1}{m}\sum_{i=1}^m \left\|\mathbf{z}_i - \mathrm{sg}[\mathbf{c}_i]\right\|_2^2}_{\text{commitment loss}}, \qquad (2)$$

where $\mathrm{sg}[\cdot]$ denotes the stop-gradient operator and $\beta_1$ is the weight.

---

[2]We use text-attributed graphs in our experiments due to the data availability, and use textual encoder to align the node features, similar to Liu et al. [45]. Despite designing node feature alignment method is crucial in GFMs, it is beyond the scope of this paper.

**Computation Tree Reconstruction.** We introduce a computation tree reconstruction task designed to enable a deep understanding of the structural and semantical attributes of computation trees [36]. We use the tree tokens to reconstruct the original computation tree, retaining general knowledge while discarding irrelevant details. Specifically, we develop three reconstruction tasks: (i) reconstructing the features of the root node $\mathcal{L}_{feat}$, (ii) reconstructing the connectivity among nodes in the computation trees $\mathcal{L}_{topo}$, and (iii) reconstructing the overall semantics of the computation trees $\mathcal{L}_{sem}$:

$$\mathcal{L}_{feat} = \frac{1}{m}\sum_{i=1}^{m}\left\|\hat{\mathbf{q}}_i^2 - \mathbf{x}_i\right\|_2^2, \qquad \mathcal{L}_{sem} = \frac{1}{m}\sum_{i=1}^{m}\Big(1 - \frac{\hat{\mathbf{q}}_i^{1\,T}\hat{\mathbf{z}}_i}{\|\hat{\mathbf{q}}_i^1\|\|\hat{\mathbf{z}}_i\|}\Big)^{\gamma}, \tag{3}$$

$$\mathcal{L}_{topo} = \sum_{(i,j)\in\mathcal{E},(i,j')\in\hat{\mathcal{E}}} -\frac{1}{|\mathcal{E}|}\log\Big(\sigma(\hat{\mathbf{q}}_i^{3\,T}\hat{\mathbf{q}}_j^3)\Big) - \frac{1}{|\hat{\mathcal{E}}|}\log\Big(1 - \sigma(\hat{\mathbf{q}}_i^{3\,T}\hat{\mathbf{q}}_{j'}^3)\Big) + \frac{1}{|\mathcal{E}|}\Big\|[\mathbf{q}_i^4\|\mathbf{q}_j^4] - \mathbf{e}_{ij}\Big\|_2^2,$$

where $\hat{\mathbf{z}}_i = \hat{\phi}(\mathcal{T}_i)$ represents the semantics of the original computation trees, and $\hat{\phi}$ is updated through a moving average of the tree encoder $\phi$. The quantized tree embedding $\mathbf{q}$ is projected via different decoders defined by MLP, $\hat{\mathbf{q}}^j = \delta_j(\mathbf{q})$, $\gamma$ is the scaling factor, and $\mathcal{E}$ and $\hat{\mathcal{E}}$ represent sets of existing and non-existing connections in computation trees, respectively. $\mathbf{e}_{ij}$ denotes the edge embedding between nodes $i$ and $j$. By jointly optimizing these tasks, we establish a comprehensive reconstruction objective:

$$\mathcal{L}_{tree} = \beta_2 \cdot \mathcal{L}_{feat} + \beta_3 \cdot \mathcal{L}_{sem} + \beta_4 \cdot \mathcal{L}_{topo}, \tag{4}$$

where $\beta_i$ indicates the weights of respective losses. The philosophies under these loss functions separately correspond to existing works [39, 26, 74, 86]. For example, Kipf and Welling [39] reconstruct the graph structure, aligning to the philosophy of $\mathcal{L}_{topo}$, Hou et al. [26] reconstruct node feature that is similar to $\mathcal{L}_{feat}$, and Thakoor et al. [74], Wang et al. [86] employ contrastive learning to maximize the alignment between two views, aligning to $\mathcal{L}_{sem}$. Unlike existing methods that typically focus on reconstructing a single aspect of computation trees, GFT integrates multiple facets [85] to learn a general and transferable tree vocabulary.

**Enhancing the Quality of Tree Vocabulary.** The effectiveness of GFT is correlated to the quality of the tree vocabulary, which should be both comprehensive and expressive. A comprehensive vocabulary is inclusive enough to accommodate new patterns, while an expressive vocabulary ensures that different tree tokens do not overlap in representation [50]. To enhance comprehensiveness, we augment the computation trees during pre-training, increasing the variety of observed computation trees through node feature augmentation and structural augmentation, as described by [115]. To improve expressiveness, we regularize the tree vocabulary space by intentionally increasing the distance between distinct tokens [65]. Specifically, we introduce an orthogonal regularizer designed to maintain tree tokens orthogonal to each other, effectively expanding the tree token space:

$$\mathcal{L}_{ortho} = \lambda\frac{1}{K^2}\Big\|\mathbf{C}\mathbf{C}^T - \mathbf{I}_K\Big\|_F^2, \quad \mathbf{C} = [\mathbf{c}_1,...,\mathbf{c}_K]^T \in \mathbb{R}^{K\times d'}, \tag{5}$$

where $\mathbf{c}_i$ is tree token, $\mathbf{I}_K$ is the identity matrix for $K$ dimensions, and $\|\cdot\|_F$ denotes the Frobenius norm. The orthogonal loss $\mathcal{L}_{ortho}$ is integrated with Equation 2. More analysis is in Appendix C.2.

## 3.2 Fine-tuning with Computation Tree Classification

The pre-training stage encodes general knowledge into the tree vocabulary, while the fine-tuning phase adapts this knowledge to specific tasks. This adaptation is challenging because identical patterns can have different meanings across domains and tasks [8]. For example, a triangular structure indicates stable relationships in social networks (node classification) but denotes unstable chemical properties in molecular networks (graph classification). To this end, we propose computation tree classification that utilizes the tree vocabulary to unify graph tasks as the tree-level task, ensuring the adaptation is applicable across diverse tasks and domains.

**Reformulate Graph Tasks as Computation Tree Classification.** Graph-related tasks can be represented by task-specific computation trees, as illustrated in Figure 1. Specifically, for node classification, the task-specific computation tree, denoted as $\mathcal{T}_{node} = \mathcal{T}_i$, is derived directly from the node itself, resulting in the embedding $\mathbf{z} = \phi(\mathcal{T}_i)$. For link prediction, the computation tree, $\mathcal{T}_{link} = \text{Combine}(\mathcal{T}_s, \mathcal{T}_t)$, merges the computation trees of two nodes of the edge, with the embedding

$\mathbf{z} = \text{mean}(\phi(\mathcal{T}_s), \phi(\mathcal{T}_t))$. For graph classification, the task-specific computation tree $\mathcal{T}_{graph} = \text{Combine}(\{\mathcal{T}_v\}_{v \in \mathcal{V}})$ integrates the computation trees of all nodes within the graph, and computes the embedding as $\mathbf{z} = \text{mean}(\{\phi(\mathcal{T}_v)\}_{v \in \mathcal{V}})$. Subsequently, the embeddings of these task-specific trees are used to query the tree vocabulary and then make predictions, adapting the general knowledge encoded in the vocabulary to various tasks and domains.

**Prototype Classifier.** The prototype classifier $f_{proto}$ constructs class prototypes using tokens from the tree vocabulary. Given a set of task-specific computation trees $\{(\mathcal{T}_i, y_i)\}_{i=1}^n$ with $|C|$ classes, we employ the pre-trained GNN encoder $\phi$ to generate tree embeddings $\mathcal{Z} = \{\mathbf{z}_i\}_{i=1}^n$. These embeddings are then used to query the tree vocabulary and produce quantized embeddings $\mathcal{Q} = \{\mathbf{q}_i\}_{i=1}^n$. Subsequently, we construct a class-wise memory bank $\mathbb{S} = \{\mathbb{S}^1, ..., \mathbb{S}^{|C|}\}$, where $\mathbb{S}^k = \{\mathbf{q}_i \in \mathcal{Q} | y_i = k\}$, to store tree tokens of the same class. The memory bank typically includes all instances from the training set. From this, we derive a set of prototypes for each class $\{\mathbf{p}_k\}_{k=1}^{|C|}$, calculated as $\mathbf{p}_k = (1/|\mathbb{S}^k|) \sum_{\mathbf{q}_i \in \mathbb{S}^k} \mathbf{q}_i$. These prototypes are then used for predictions:

$$p(y = k|\mathbf{z}) = \frac{\exp(-\text{sim}(\mathbf{z}, \mathbf{p}_k)/\tau)}{\sum_c \exp(-\text{sim}(\mathbf{z}, \mathbf{p}_c)/\tau)}, \tag{6}$$

where $\text{sim}(\cdot)$ denotes the cosine distance and $\tau$ is a temperature scaling factor. We optimize the cross-entropy loss between the classifier's output and the ground truth to update the encoder $\phi$.

**Linear Classifier.** Different from the prototype classifier, which utilizes class prototypes to adapt to target tasks, the linear classifier $f_{lin}$ directly applies the knowledge encoded in each tree token. Specifically, given a task-specific computation tree $\mathcal{T}_i$, we use the encoder to generate tree embeddings $\mathbf{z}_i$ and then query the tree vocabulary to retrieve $\mathbf{q}_i$. These embeddings are used for predictions as:

$$p(y = k|\mathbf{z}) = \frac{\exp(\text{lin}^k(\mathbf{q})/\tau)}{\sum_c \exp(\text{lin}^c(\mathbf{q})/\tau)}, \tag{7}$$

We optimize the cross-entropy loss between the prediction $f_{lin}(\mathbf{z})$ and the ground truth to update the parameters of the encoder and the linear classifier. During inference, predictions from both the prototype and linear classifiers are combined to form the final output. It is important to note that the tree vocabulary remains fixed during fine-tuning to preserve the integrity of the encoded knowledge.

### 3.3 Additional Analysis

**Tree Vocabulary Learns Generalizable Tokens.** Learning tree vocabulary via VQ involves clustering within the embedding space of computation trees, utilizing a margin-aware classifier [14] that assigns each computation tree to a specific cluster. Assuming that each computation tree $\mathcal{T}$ is associated with an underlying clustering label $y$, and that each pair $(\mathcal{T}_i, y_i)$ is sampled from the distribution $\mathcal{P}_\mathcal{T}$, we derive the following theorem:

**Theorem 3.1.** *Given a set of computation trees $\{(\mathcal{T}_i, y_i)\}_{i=1}^n$ sampled from the distribution $\mathcal{P}_\mathcal{T}$, the VQ process functions as a margin-aware prototype classifier $f$ that predicts the class of computation trees via a distance measure. The risk $\mathcal{R}(f)$ of classifier $f$ can be bounded with probability $1 - \delta$:*

$$\mathcal{R}(f) \leq \hat{\mathcal{R}}(f) + \frac{20 \cdot \mathcal{C} \cdot p(p-1) \cdot \sqrt{n}}{\rho \cdot n} + \sqrt{\frac{\ln(2/\delta)}{2n}}, \tag{8}$$

*where $\hat{\mathcal{R}}(f)$ is the empirical risk, $p$ denotes the number of tokens, $\mathcal{C}$ is a constant, and $\rho$ acts as the margin, serving as a penalty factor in evaluating the distance between computation trees and tokens.*

*Remark* 3.2. The generalizability of tokens within the vocabulary highly correlates to the margin $\rho$, the number of observed computation trees $n$, and the number of tokens $p$. (i) A larger margin $\rho$ results in a tighter bound by ensuring higher inter-cluster distances and lower intra-cluster distances. This supports the use of an orthogonal regularizer (Equation 5) that explicitly pushes tokens apart, enhancing cluster distinction. (ii) An increased number of observed computation trees $n$ leads to a tighter generalization bound, which supports the use of augmentations to increase the diversity of computation trees. (iii) More tokens $p$ may loose the upper bound of the generalization error, potentially due to a higher risk of overfitting. This aligns with our experimental findings that more tokens do not necessarily lead to improved performance (Section 4.4).

**Tree Vocabulary Mitigates Negative Transfer.** Negative Transfer (NT) occurs when the pre-training process degrades model performance on a target task. This issue often results from misalignment between the pre-training and fine-tuning tasks [89, 8]. Following the approach in [89], we characterize the NT gap, $\mathcal{R}(S,T) - \mathcal{R}(\emptyset,T)$, as the risk gap on task $T$ with ($\mathcal{R}(S,T)$) and without ($\mathcal{R}(\emptyset,T)$) pre-training on task $S$, where a smaller NT gap indicates improved transferability. As illustrated in Figure 5, employing the learned tree vocabulary to align the tree reconstruction task in pre-training and tree classification task in fine-tuning can significantly mitigate negative transfer.

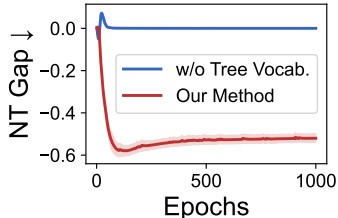

Figure 5: Negative transfer gap on `Cora` in node classification.

**Complexity Analysis.** A comprehensive complexity analysis of GFT is provided in Appendix B. Notably, GFT employs a single GNN to decompose and encode computation trees, taking $\mathcal{O}(L \cdot (|\mathcal{E}| \cdot d + |\mathcal{V}| \cdot d^2))$. In contrast, subgraph-based GFMs [30, 45] require the explicit extraction of subgraphs for each node, taking additional $\mathcal{O}(|\mathcal{V}|^3)$ using adjacency matrix-based BFS. This contrast highlights the efficiency of using computation trees as transferable patterns in terms of time complexity. More discussions are in Appendix C.

## 4 Experiments

### 4.1 Experimental Setting

We employ cross-domain and cross-task graph datasets to evaluate the effectiveness of GFT. For node-level tasks, we utilize citation networks such as `Cora`, `PubMed`, `Arxiv`, and the web link network `WikiCS`. For edge-level tasks, we include two Knowledge Graphs (KGs), `WN18RR` and `FB15K237`. For graph-level tasks, we use molecule networks, including `HIV`, `PCBA`, and `ChEMBL`. All preprocessing steps follow [45]. We take various baselines, encompassing MLP, supervised GNNs such as GCN [40], GAT [78], GIN [94], and self-supervised methods like BGRL [74], GraphMAE [26], GIANT [11], and GFMs including Prodigy [30] and OFA [45]. We replicate each experiment ten times and report the average performance to minimize the influence of randomness. Further details on experimental settings are available in Appendix F.

### 4.2 Effectiveness on Cross-Domain and Cross-Task Datasets

**Pre-training and Fine-tuning.** Table 2 demonstrates the model performance across cross-domain and cross-task datasets in pre-training and fine-tuning setting. We evaluate the effectiveness of graph foundation models [30, 45] in the following few-shot setting due their distinctive training mechanisms, such as in-context pre-training [30] and fully supervised training [45]. For supervised baselines, models are trained directly on the target graph; for self-supervised methods, we pre-train across all datasets before adapting to the specific target graph. Our approach demonstrates a substantial performance improvement, exceeding the best baseline by an average of over 6%. Specifically, our method outperforms the best baseline by 2% across three datasets and by 5% across another three datasets. This underscores the effectiveness of using computation trees as transferable patterns.

**Few-shot Learning** Table 3 presents the few-shot learning performance of GFT compared to self-supervised methods [74, 26, 11] and graph foundation models [30, 45, 25]. We randomly select $k$ samples per way from the training set for fine-tuning[3]. This method is similar to Prodigy [30], and is much more label-efficient than OFA [45] with supervised pre-training. Despite the extremely limited labels for fine-tuning, GFT significantly surpasses existing methods, showing the fast adaptation capability. Appendix H shows more fine-tuning instances can significantly improve performance.

### 4.3 Transferability

Table 5 shows the impact of different pre-training datasets under the pre-training and fine-tuning setting, where comprehensive results (including the following ablation studies) are available in Appendix I. The performance for specific tasks (node-, link-, graph-level) represent the average across all involved datasets. We examine three scenarios with distinct pre-training datasets: (i) all

---

[3]`Cora` & `WN18RR`: 1; `Arxiv`: 5; `HIV` & `PCBA`: 20; `FB15K237`: 30.

Table 2: Model performance in pre-training and fine-tuning setting. **Bold** and underline highlight the best and sub-best performance, and * and ‡ denote a 2% and 5% improvement over the best baseline. The model performance with standard deviation is in Appendix G.

| Method | Node Classification | | | | Link Classification | | Graph Classification | | |
|---|---|---|---|---|---|---|---|---|---|
| | Cora | PubMed | Wiki-CS | Arxiv | WN18RR | FB15K237 | HIV | PCBA | *Avg.* |
| Linear | 58.03 | 68.66 | 70.36 | 66.50 | 78.50 | 87.39 | 66.37 | 72.30 | 71.01 |
| GCN [40] | 75.65 | 75.61 | 75.28 | 71.40 | 73.79 | 82.22 | 64.84 | 71.32 | 73.76 |
| GAT [78] | 76.24 | 74.86 | 76.78 | 70.87 | 80.16 | 88.93 | 65.54 | 70.12 | 75.44 |
| GIN [94] | 73.59 | 69.51 | 49.77 | 65.05 | 74.02 | 83.21 | 66.86 | 72.69 | 69.34 |
| DGI [79] | 72.10 | 73.13 | 75.32 | 69.15 | 75.75 | 81.34 | 59.62 | 63.31 | 71.22 |
| BGRL [74] | 71.20 | 75.29 | 76.53 | 71.19 | 75.44 | 80.66 | 63.95 | 67.09 | 72.67 |
| GraphMAE [26] | 73.10 | 74.32 | 77.61 | 70.90 | 78.99 | 85.30 | 61.04 | 63.30 | 73.07 |
| GIANT [11] | 75.13 | 72.31 | 76.56 | 70.10 | 84.36 | 87.45 | 65.44 | 61.49 | 74.11 |
| GFT | **78.62**\* | **77.19**\* | **79.39**\* | **71.93** | **91.91**‡ | **89.72** | **72.67**‡ | **77.90**‡ | **79.92**‡ |

Table 3: Few-shot learning performance. Additional results with more baselines are in Appendix H.

| Method | Arxiv - 40 way | | | Arxiv - 5 way | | | FB15K237 - 40 way | | | FB15K237 - 10 way | | |
|---|---|---|---|---|---|---|---|---|---|---|---|---|
| | 5-shot | 3-shot | 1-shot | 5-shot | 3-shot | 1-shot | 5-shot | 3-shot | 1-shot | 5-shot | 3-shot | 1-shot |
| BGRL [74] | - | 17.98 | - | - | 48.43 | - | - | 29.24 | - | - | 67.23 | - |
| GraphMAE [26] | - | 19.12 | - | - | 49.24 | - | - | 32.07 | - | - | 69.75 | - |
| GIANT [11] | - | 20.12 | - | - | 54.33 | - | - | 52.63 | - | - | 77.21 | - |
| Prodigy [30] | 25.51 | 23.69 | 21.44 | 61.09 | 58.64 | 48.23 | 62.03 | 59.58 | 54.30 | 84.30 | 79.61 | 66.10 |
| OFA [45] | 24.01 | 22.13 | 21.34 | 59.92 | 58.68 | 52.80 | 66.51 | 65.76 | 63.48 | 83.64 | 83.14 | 83.46 |
| GFT | **36.29**‡ | **34.36**‡ | **26.49**‡ | **68.00**‡ | **66.00**‡ | **58.20**‡ | **75.01**‡ | **74.56**‡ | **74.97**‡ | **89.13**‡ | **88.53**‡ | **88.07**‡ |

| Method | WN18RR 10-way | | | Cora 5-way | | | HIV 2-way | | | PCBA 2-way | | |
|---|---|---|---|---|---|---|---|---|---|---|---|---|
| | 5-shot | 3-shot | 1-shot | 5-shot | 3-shot | 1-shot | 10-shot | 5-shot | 1-shot | 10-shot | 5-shot | 1-shot |
| OFA [45] | 32.64 | 30.56 | 25.82 | 42.28 | 31.28 | 23.68 | 54.36 | 57.56 | 57.17 | 54.58 | 54.80 | 54.92 |
| GFT | **35.50**‡ | **35.50**‡ | **35.33**‡ | **52.30**‡ | **51.47**‡ | **49.80**‡ | **58.67**‡ | **58.78**\* | **59.94**\* | **59.34**‡ | **59.34**‡ | **55.88** |

datasets, (ii) only the target dataset, and (iii) datasets excluding the target dataset. These variants are compared against GAT and GIANT, which represent the best supervised and self-supervised baselines, respectively. Notably, GFT consistently outperforms all baselines, regardless of the pre-training dataset utilized. Interestingly, performance improves when using datasets excluding the target dataset compared to pre-training solely on the target dataset. We hypothesize that the computation trees from the non-target datasets provide sufficient information to facilitate the learning of a transferable tree vocabulary, thereby promoting positive transfer.

We further evaluate the impact of various combinations of pre-training datasets on the target tasks, as depicted in Figure 7. For pre-training, we select FB15K237, Arxiv, and ChEMBL, while Cora, WikiCS, WN18RR, and HIV serve as the target datasets. Our findings indicate that an increased number of pre-training datasets consistently enhances performance across all target datasets. However, for existing GFMs, transferability closely correlates with the selection of pre-training datasets, with more datasets sometimes leading to negative transfer [25, 43]. This observation underscores the adaptability of using computation trees as transferable patterns in graph vocabulary.

## 4.4 Ablation Study

**Tree Reconstruction and Classification.** Table 4 shows the impact of various reconstruction tasks in pre-training and tree classifiers in fine-tuning. All reconstruction tasks enhance model performance compared to models without pre-training. Notably, semantic reconstruction is most effective for node-level and graph-level tasks due to its comprehensive consideration of node features and graph structures. Feature reconstruction is particularly beneficial for link-level tasks, as it preserves the original node semantics, which are crucial for KGs. The optimal performance is achieved when three tasks are jointly optimized, aligning with findings in Ju et al. [36]. Similarly, the combination of prototype and linear classifiers in tree classification leads to superior performance. Furthermore, removing strategies designed to enhance the quality of the tree vocabulary results in model degradation across all settings (Appendix I.3).

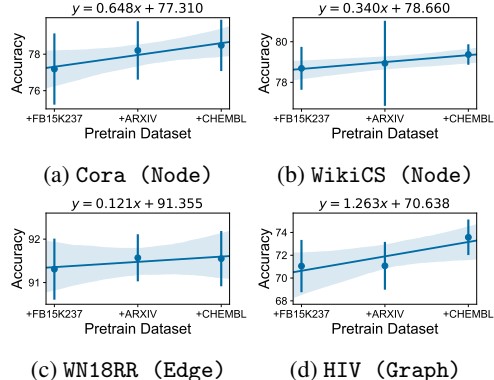

(a) `Cora (Node)`  (b) `WikiCS (Node)`

(c) `WN18RR (Edge)`  (d) `HIV (Graph)`

Figure 7: GFT consistently improves model performance with more pre-training datasets.

Table 4: Ablation on tree reconstruction (above) and tree classification (bottom).

|  | Node | Link | Graph | Avg. |
|---|---|---|---|---|
| *Different Pre-training Tasks* | | | | |
| n/a | 72.52 | 47.30 | 73.83 | 66.54 |
| w. $\mathcal{L}_{sem}$ | 76.25 | 90.39 | 74.99 | 79.47 |
| w. $\mathcal{L}_{feat}$ | 75.85 | 90.42 | 74.42 | 79.13 |
| w. $\mathcal{L}_{topo}$ | 75.50 | 90.28 | 74.57 | 78.96 |
| *Different Fine-tuning Classifiers* | | | | |
| w. $f_{proto}$ | 76.64 | 38.71 | 59.89 | 62.97 |
| w. $f_{lin}$ | 75.51 | 88.99 | 72.21 | 78.06 |
| GFT | **76.78** | **90.82** | **75.29** | **79.92** |

Table 5: The impact of pre-training datasets.

|  | Node | Link | Graph | Avg. |
|---|---|---|---|---|
| GAT [78] | 74.69 | 84.55 | 67.83 | 75.44 |
| GIANT [11] | 73.53 | 85.91 | 63.47 | 74.11 |
| *Different Pre-training Datasets* | | | | |
| All Datasets | **76.78** | **90.82** | **75.29** | **79.92** |
| Target Dataset | 76.12 | 90.67 | 74.08 | 79.25 |
| Remaining Datasets | 75.94 | 90.71 | 74.86 | 79.36 |

Table 6: The impact of tree vocabulary.

|  | Node | Link | Graph | Avg. |
|---|---|---|---|---|
| *Different Vocabulary Size* | | | | |
| # Tokens = 128 | 76.78 | 90.82 | **75.29** | 79.92 |
| # Tokens = 256 | 76.71 | **90.86** | 75.17 | 79.86 |
| # Tokens = 512 | **76.94** | **90.86** | 75.21 | **79.99** |
| *Without Vocabulary* | | | | |
| w/o. Vocab. | 75.90 | 86.70 | 69.17 | 76.91 |

**Tree Vocabulary.** Table 6 shows the importance of the vocabulary, where the use of vocabulary significantly enhances model performance, particularly in link- and graph-level tasks, which aligns to the theoretical analysis that the tree vocabulary improves generalization. However, we observe that increasing the number of tokens in the vocabulary does not necessarily enhance model performance; indeed, the improvements are often marginal.

## 5  Conclusion

**Conclusion.** In this paper, we rethink the transferable pattern in graphs as computation trees and validate their transferability both empirically and theoretically. Building on this insight, we propose a cross-domain and cross-task GFM named GFT. This model leverages computation tree reconstruction to acquire general graph knowledge from cross-domain datasets and uses computation tree classification to facilitate adaptation to various target tasks. In future work, we aim to explore its capabilities for in-context learning and zero-shot learning.

**Limitations.** In this paper, we focus primarily on message-passing GNNs, as message-passing can be naturally unrolled as a tree-like structure. However, our analysis excludes graph transformers and expressive GNNs with specialized computational architectures. We plan to extend our analysis to understand the transferable patterns of these advanced learning algorithms in future work. Additionally, message-passing GNNs may lack the expressiveness needed to address isomorphism problems in graphs. One can apply advanced techniques [105] to handle link isomorphism and use advanced expressive GNNs [103] to tackle graph isomorphism. Moreover, the deployment of GFT in real-world applications may encounter efficiency issues, which can be mitigated by techniques like [106, 88].

**Boarder Impact.** The proposed GFT is a cross-domain and cross-task graph foundation model, designed for rapid adaptation to target tasks with extremely limited labels. We wish our research can support applications where label acquisition is challenging and model training is time-consuming, such as in molecular discovery and financial fraud detection.

## Acknowledgments

This work was partially supported by the NSF under grants IIS-2321504, IIS-2334193, IIS-2340346, IIS-2203262, IIS-2217239, CNS-2426514, CNS-2203261, and CMMI-2146076. Any opinions, findings, and conclusions or recommendations expressed in this material are those of the authors and do not necessarily reflect the views of the sponsors.

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

## Appendix: Table of Content

# A    More Related Work

**Transferability of GNNs.**    Early studies on the transferability of GNNs are based on two main approaches. The first approach utilizes graphon theory. Ruiz et al. [62] derive a bound between the embeddings of two graphs sampled from the same graphon. However, identifying a common graphon in many real-world graphs is often unfeasible, limiting the direct application of this theorem in the design of graph foundation models. Cao et al. [8] employ the graphon theory to analyze transferability in pre-training and fine-tuning setting. They fit pre-trained graphs into a graphon space, ensuring transferability if the target graph can be generated within this space. Like Ruiz et al. [62], the challenge lies in acquiring sufficient data to adequately represent the graphon space for graph foundation model. Following this, Sun et al. [69] designed a fine-tuning method based on graphon theory. Although these studies consider graphon as a transferable pattern in graphs, the assumption is challenging to satisfy in cross-domain real-world graphs. Furthermore, graphon theory is generally limited to single-task applications, making it difficult to identify a shared graphon across node-, link-, and graph-level tasks. The send approach examines transferability through subgraphs [114] or graph spectrum [41, 42]. Specifically, Levie et al. [41] analyze transferability from the perspective of stability, positing that effective transferability minimizes the impact of small perturbations. Similarly, Levie et al. [42] explore transferability through stability, demonstrating that transfer between graphs is feasible when they discretize the same underlying space in a generic sense. Zhu et al. [114] focus on transferability through ego-graphs, showing a higher similar ego-graph distribution leads to better transferability. In contrast to these two approaches, we treat computation trees as transferable patterns on graphs, and conduct both empirically and theoretically analysis to show their transferability. Additionally, we develop a graph foundation model that utilizes these computation tree patterns.

**Generalization of GNNs.**    Generalization is a closely related topic to transferability. For instance, Scarselli et al. [63] pioneer the analysis of GNNs' VC-dimension, focusing solely on the number of nodes. Garg et al. [20] leverage Rademacher complexity to evaluate GNN generalization through computation tree perspectives. Furthermore, the Rademacher complexity has been extended to transductive settings by Esser et al. [16] and Cong et al. [13]. Under the PAC-Bayesian framework, Liao et al. [44] offer a tighter generalization bound for GNNs compared to Garg et al. [20], and Ju et al. [35] further improve the bound. Additionally, Sun et al. [66] investigate these bounds through the lens of graph topology. Stability is another lens through which generalization is examined, with Verma and Zhang [80] focusing on 1-layer GNNs and linking generalization to the largest absolute eigenvalue of the graph convolution filter. Tang and Liu [72] further establish bounds for transductive node classification, highlighting the significance of graph structural information for different GNN architectures.

**GNN-based Graph Foundation Models.**    Developing graph foundation models involves two primary steps: (i) unifying the task space and (ii) unifying the domain space. Several studies focus on aligning the task space. Qiu et al. [59] introduce a self-supervised model that empirically demonstrates the transferability of subgraphs across tasks. Sun et al. [67], Liu et al. [47] pinpoint the task gap between pre-training and fine-tuning as the primary performance bottleneck, addressing it through link prediction to unify these tasks. Yan et al. [95] further adapt this model to an inductive setting, where the pre-training and fine-tuning graphs differ, proposing methods to bridge both the graph signal and structure gaps. Yu et al. [99] implement multi-task pre-training to support various downstream tasks. Sun et al. [68] employ subgraphs as fundamental transferable patterns, integrating node-, link-, and graph-level tasks into a unified subgraph-level task. Instead of extensive model fine-tuning, they incorporate a learnable subgraph into the original graph. Other research focuses on aligning the domain space. Li et al. [43] introduce a zero-shot graph learning framework for cross-domain node classification, leveraging LLMs to unify node features across different graphs. In a similar vein, Zhao et al. [108] propose a graph prompting method for cross-domain classification, utilizing singular value decomposition to align the feature space across various graphs. However, all of these methods are generally limited to single tasks or domains, and do not effectively address the complexities of datasets that span multiple domains and tasks.

To this end, Huang et al. [30] introduce a graph foundation model that utilizes LLMs to align the feature space of graphs and utilizes in-context learning to facilitate applications in node-level and link-level tasks. In particular, they extract the subgraphs for different tasks and conduct subgraph classification. However, this approach necessitates that the pre-training and fine-tuning tasks be

identical due to the specialized in-context pre-training strategy. To address this constraint, Liu et al. [45] use LLMs to align the feature spaces of cross-domain graph datasets and introduce a prompt graph to align various tasks on graphs. Xia et al. [93] leverages a graph tokenizer to convert the graph into sequence and propose to use transformer to handle such information. Following, He and Hooi [25] concurrently train GNNs and LLMs to enhance performance further. Despite their empirical success, subgraph-based graph foundation models face challenges due to the GNNs' limitations in encoding substructures within subgraphs. Differently, we rethink computation trees as transferable patterns and propose a new graph foundation model based on that.

**LLM-based Graph Foundation Models.**  LLMs present a promising avenue for the development of graph foundation models due to their ability to unify the output of various graph tasks [17, 22, 81]. While GNNs require task-specific adjustments for model training, LLMs can accommodate a wide range of questions and generate appropriate answers. The primary challenge, though, lies in effectively translating graph structures into a natural language format that LLMs can comprehend. Current efforts in this domain focus on two main approaches. The first is to use natural language to describe the graph structure, such as what the nodes are and which pairs of nodes are connected [17]. Such methods can be further enhanced with extra embedding [9] or prompt engineer techniques to enhance the understanding of LLM. For example, Guo et al. [22] employ a self-prompting methods to utilize the context generated by LLM as input context; Chai et al. [9] used Build-a-Graph and Algorithmic prompting to faciliate LLM understanding. Zhao et al. [109] map a graph into tree-like tokens for designing LLM prompt, further enhancing learning capability. Additionally, another line of works [75, 73] follow Visual Language Models (VLMs) to process the graph into embeddings by GNNs first and then employ LLM as a translator to decode the graph embedding.

It is noted that prior research such as Wang et al. [81], Huang et al. [29] indicate that LLMs can indeed capture structural information from graphs, which enhances their performance on downstream tasks. However, while LLMs show promise in basic graph reasoning tasks like connectivity checks and cycle detection, they struggle with complex graph patterns in graph learning tasks such as node and graph classification. Moreover, there is limited research on cross-domain graph foundation models, largely due to the diverse patterns and distributions of graphs across different domains. This underscores the importance of our work in identifying transferable patterns within graphs to pave the way for future advancements in graph foundation models.

**Computation Tree.**  The computation tree, or more broadly, the subtree, is a fundamental structure on graphs [20]. It serves to (i) enhance the performance of existing GNNs and to (ii) measure graph similarity. Several studies [34, 18, 70, 84] employ tree decomposition to develop advanced GNNs. Specifically, Jin et al. [34] treat the joint tree as complementary to the original graphs, while Fey et al. [18] introduce inter-modality message passing between joint trees and the original graphs. Talak et al. [70] construct an H-tree by organizing nodes and subgraphs hierarchically, developing a neural tree model capable of approximating any probability distribution on a probabilistic graphical model. Wang and Derr [84] propose a more efficient tree decomposition algorithm by separating the model layer and tree construction. Furthermore, Huang et al. [31] investigate the significance of trees in learning node representations and design a framework to identify the most crucial trees in a graph. Additionally, Nikolentzos et al. [54] design a hyperbolic learning framework to utilize the computation tree structure in creating expressive GNNs. Bai et al. [2] adapt the computation tree concept to knowledge graphs by optimizing the solution in query computation trees. Another avenue of research utilizes computation tree distributions to measure graph similarity. Notably, Shervashidze et al. [64] introduce the WL subtree kernel to measure discrepancies between graphs based on subtree structures. Chuang and Jegelka [12] employs optimal transport to propose the tree mover's distance, estimating distribution shifts between graphs. Wu et al. [91] utilizes a hierarchical WL subtree kernel to assess graph discrepancies and derive a generalization bound for cross-domain classification.

Different from these approaches, our work rethink the role of computation trees. We consider computation tree as a transferable pattern on graphs and both empirically and theoretically validate its transferability, thereby expanding the analysis for computation trees in graph learning.

# B Complexity Analysis

## B.1 Computation Tree Decomposition and Encoding

The decomposition and encoding of computation trees can be jointly finished by message-passing GNNs. Specifically, the learning process in message passing GNNs involves (i) extracting computation trees for each node and (ii) updating node embeddings within these trees from bottom to top. Utilizing a GraphSAGE-like architecture, as detailed in Appendix F, each layer's learning comprises both aggregation and updating operations. We will analyze these two operations in the following.

Aggregation is an edge-wise operation that propagates messages from neighboring nodes to the target node. Consequently, this operation's computational complexity is linear to the number of edges in the graph $\mathcal{G} = (\mathcal{V}, \mathcal{E})$, expressed as $\mathcal{O}(|\mathcal{E}| \cdot d)$, where $d$ is the embedding dimension. The update process, on the other hand, is a node-wise operation that updates the state of each node based on aggregated messages through a neural network. Therefore, its time complexity is $\mathcal{O}(|\mathcal{V}| \cdot d^2)$, as the complexity of the neural network operations per node is $\mathcal{O}(d^2)$. By integrating both aggregation and update processes in each layer, the overall complexity of our model is $\mathcal{O}(L \cdot (|\mathcal{E}| \cdot d + |\mathcal{V}| \cdot d^2))$.

For node-level tasks, we directly use the embeddings of computation trees of the target node, which incurs a constant time complexity of $\mathcal{O}(1)$. For link-level and graph-level tasks, we apply a non-parametric pooling function to aggregate subtree embeddings into a computation tree embedding, also with a time complexity of $\mathcal{O}(1)$.

## B.2 Subgraph Extraction

Current graph foundation models [30, 45] treat subgraphs as transferable patterns and explicitly extract subgraphs for each node. For a given graph $\mathcal{G} = (\mathcal{V}, \mathcal{E})$, the extraction of ego-graph for all nodes results in a computational cost of $\mathcal{O}(|\mathcal{V}|^3)$ when using an adjacency matrix for the BFS algorithm.

## B.3 Vector Quantization

Vector quantization assigns each instance to the nearest token in the vocabulary. This process involves measuring the distance between the instance and every token, then selecting the token with the minimum distance as the quantized embedding. Assuming there are $K$ tokens, the complexity of distance measurement (no matter Euclidean or Cosine) is $\mathcal{O}(K \times d)$ per instance. Then, determining the shortest distance from $K$ measured distances can be achieved in $\mathcal{O}(K)$, and replacing the original instance embedding with the selected token requires $\mathcal{O}(1)$. Therefore, the predominant computational cost is distance measurement, leading to an overall complexity of $\mathcal{O}(|\mathcal{T}| \cdot K \cdot d)$, where $|\mathcal{T}|$ represents the number of computation trees.

## B.4 Tree Reconstruction

The computation tree reconstruction comprises three main tasks: feature reconstruction, semantic reconstruction, and topology reconstruction. Feature reconstruction utilizes a neural network-based decoder with a complexity of $\mathcal{O}(d^2)$ and a mean squared error (MSE) loss of $\mathcal{O}(|\mathcal{T}|)$, resulting in a total complexity of $\mathcal{O}(d^2 + |\mathcal{T}|)$. Topology reconstruction focuses on edge information and takes a computational cost of $\mathcal{O}(|\mathcal{E}| \cdot d)$. Semantic reconstruction involves an additional GNN and a distance measurement, leading to a complexity of $\mathcal{O}(L \cdot (|\mathcal{E}| \cdot d + |\mathcal{V}| \cdot d^2) + |\mathcal{T}| \cdot d)$. Consequently, the overall computational complexity is approximated as $\mathcal{O}(L \cdot (|\mathcal{E}| \cdot d + |\mathcal{V}| \cdot d^2) + |\mathcal{T}| \cdot d)$.

## B.5 Tree Classification

The computation tree classification process employs both a prototype-based classifier and a linear classifier. The prototype-based classifier constructs prototypes from a memory bank, which incurs a complexity of $\mathcal{O}(|\mathcal{T}|)$. It then classifies instances by measuring their distances to these prototypes, resulting in a complexity of $\mathcal{O}(|\mathcal{T}| \cdot |C| \cdot d)$, where $|C|$ represents the number of classes. On the other hand, the linear classifier incurs a complexity of $\mathcal{O}(|\mathcal{T}| \cdot d)$. Consequently, the total computational complexity can be approximated as $\mathcal{O}(|\mathcal{T}| \cdot |C| \cdot d)$.

# C   More Analysis

## C.1   The Difference Between Computation Tree and Subgraph

Our concept of computation trees is closely aligned with [12], representing tree-like patterns derived from unfolding the message passing process. Encoding the computation trees of a node is equivalent to encoding the node itself via message passing GNNs, implying that the information in computation trees can be fully learned by basic GNNs, demonstrating both learnability and efficiency in encoding computation trees. Notably, computation tree can be reinterpreted as a special pattern preserved on the ego-graph of the target node, differing from junction trees [34] or H-trees [70], which construct additional tree-like graphs to complement the original graph.

Subgraphs, on the other hand, are graph-like substructures within the original graph, such as motifs in molecule graphs. Sun et al. [68] identifies subgraphs as basic patterns across graph-related tasks and reformulates these tasks into subgraph-level tasks. For example, in node classification, they extract the ego-graph around each node and assign the label of the induced graph as the label of the center node, converting node classification into subgraph classification. This process involves (1) extracting ego-graphs around task-relevant nodes and (2) using GNNs to learn graph-level embeddings for classification. However, this extraction process introduces additional time consumption and increased memory usage for storing induced subgraphs. More importantly, the information in subgraphs is not always learnable by basic GNNs, as they cannot detect some critical substructures necessary for learning graph-level embeddings [10, 103], reducing the feasibility of using subgraphs to define graph vocabularies.

We provide empirical analysis for better understanding. Efficiency analysis is presented in Figure 8. Subgraphs generally incur an extra 1/3 time consumption compared to computation trees and encounter out-of-memory errors when batch size exceeds 2048, compared to 8192 for computation trees. The performance comparison is shown in Table 7, where the subgraph version (GFT - Subgraph) performs worse than the computation tree version (GFT). We use GAT and GraphMAE as additional baselines and apply linear classifiers on all models for a fair comparison.

Table 7: The comparison between computation trees and subgraphs.

|  | Node | Link | Graph | Avg. |
|---|---|---|---|---|
| GAT | 74.69 | 84.55 | 67.83 | 75.44 |
| GraphMAE | 73.98 | 82.15 | 62.17 | 73.07 |
| GFT - Subgraph | 74.23 | 86.49 | 67.89 | 76.13 |
| GFT - Tree | 75.51 | 88.99 | 72.21 | 78.06 |

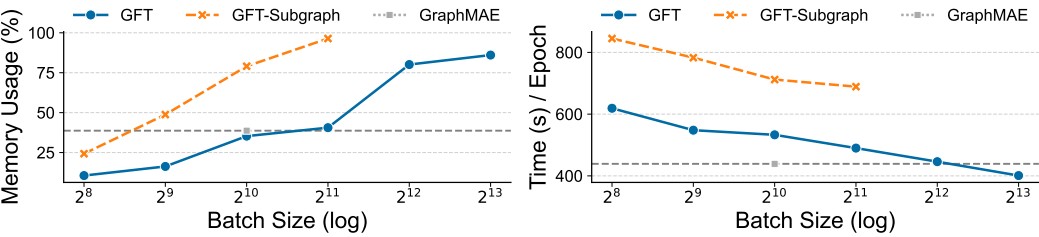

(a) The memory allocation in an A40 (48GB) GPU.          (b) The time consumption per epoch.

Figure 8: The efficiency analysis between computation trees and subgraphs. Our GFT is based on the computation trees and we further replace the computation trees with subgraphs called GFT-Subgraph. We compare their memory usage (a) and time consumption (b) during pretraining. With the increase of batch sizes, Subgraph-based GFT encounters out-of-memory, yet computation tree-based GFT can still fit in the GPU.

## C.2 Preventing Vocabulary Collapse

Another challenge in developing a robust discrete vocabulary is known as vocabulary collapse (or codebook collapse in VQ). The commitment loss (Equation 2) effectively prevents this issue by aligning the quantized tree embeddings with the token space [77]. Furthermore, we have empirically discovered that using Euclidean distance to query tree tokens leads to vocabulary collapse. Consequently, we have switched to Cosine distance to enforce querying within a hyper-sphere space, thereby enhancing training stability [97]. Alternatively, other techniques such as expiring stale codes [101] or affine re-parameterization [32] (not evaluated in this paper) can also be employed to mitigate this problem.

## C.3 Scaling to Large-scale Graphs

Due to the emergence of large-scale graphs [27, 28], efficient training often requires the use of mini-batches. We facilitate mini-batch training through subgraph sampling. In pre-training, we employ basic subgraph sampling techniques [24] to extract a smaller graph from the original graph and then extract computation trees for each node within this subgraph. This method serves as an additional topology augmentation, further enhancing the diversity of computation trees through re-sampling. In the fine-tuning phase, subgraph sampling remains effective for the linear classifier, as it directly processes the computation tree. However, the prototype-based classifier, which requires the aggregation of instances with identical labels to form class prototypes, faces efficiency challenges in this mini-batch training setting. To address this, we randomly sample a small subset of the training set for each class to construct the memory bank $\mathbb{S}$. Based on our empirical observations, a limited number of samples per class suffices to achieve desirable performance.

## C.4 Discussion on Homophily and Heterophily

Homophily and heterophily [49] are both critical properties for node-level tasks. The primary distinction between these types of graphs is that identical connectivity patterns can indicate different meanings. We consider our model is also effective for heterophily graphs. Although we only evaluate the performance of GFT on homophily graphs (Cora, PubMed, WikiCS, Arxiv), two considerations support its applicability to heterophily graphs: (i) The analysis of computation tree transferability shows that, similar to homophily, higher computation tree similarity in heterophily graphs correlates with enhanced transferability, matching the principle of our GFT. (ii) Our proposed computation tree classification in fine-tuning can adaptively reinterpret the patterns encoded in the tree vocabulary across various tasks. We will leave the experiments on heterophily graphs in the future work.

## C.5 Comparison to VQGraph

The major connection between GFT and VQGraph [96] is the usage of vector quantization in learning a discrete vocabulary for downstream tasks. However, there are four major differences between GFT and VQGraph. (1) Model Objective: GFT focuses on building a general task reasoner, but VQGraph aims to train a structure-aware MLP for efficient inference. (2) Pretrain Dataset: GFT is pre-trained on cross-domain and cross-task datasets to acquire transferable patterns among graphs, but VQGraph is pre-trained on a single dataset to better capture the structural information. (3) Usage of Tokens: GFT treats tokens as specific transferable patterns, using them directly to build classifiers. VQGraph, on the other hand, treats tokens as external structural knowledge to complement the training of MLP classifiers. (4) Downstream Tasks: GFT can be applied to various graph-related tasks with different settings like few-shot and zero-shot learning. VQGraph is designed for node classification with a basic pre-training and fine-tuning setting.

## C.6 Comparison to LLM-based Methods

Recent researches [17, 22, 81] utilize LLMs to reformulate graph-related tasks as question answering, transforming graph datasets into sentence structures and leveraging the inference capabilities of LLMs to implicitly infer structural knowledge from the original graphs. This approach exploits the transferable patterns in the word vocabulary of LLMs to reinterpret the transferable patterns on graphs. The main challenges include (i) aligning the transformed graphs (sentences) with the word vocabulary of LLMs, and (ii) employing LLMs to infer essential structural knowledge for

graph-structured data. Due to these challenges, existing methods often fall short in handling graph datasets with LLMs, resulting in inconsistent performance. Unlike these approaches, which entirely abandon GNNs, we utilize a GNN as an encoder to analyze transferable patterns on graphs. We consider these LLM-based approaches as complementary to our work.

### C.7 Comparison to Subgraph-based GFMs

Several studies [30, 45] identify subgraphs as transferable patterns across graphs, unifying graph-related tasks into subgraph classification tasks and explicitly extracting subgraphs for classification. However, this extraction process incurs additional time and memory costs due to overlapping nodes in the extracted subgraphs. Moreover, other research [20, 103] suggests that certain substructure or motif patterns within subgraphs are not learnable by basic message-passing GNNs. Unlike these methods, our GFT treats computation trees as transferable patterns, offering advantages over these GFMs in both respects. Firstly, GFT does not require the explicit extraction and encoding of computation trees, instead employing message passing to inherently processes computation trees rooted at all nodes, ensuring efficiency in both time and memory. Furthermore, the computation tree can be seen a unique subgraph structure, which is fully learnable by GNNs without information loss.

Table 8: Comparison of number of parameters across different models

| Model | # Params |
|---|---|
| Prodigy | 2M |
| OFA | 29M |
| UniGraph | 180M |
| GFT | 7M |

In addition, we also compare the number of parameters of these GFMs in Table 8. Considering the number of parameters, Prodigy [30] has 2 million parameters, while OFA [45] has 29 million since the use of more GNN layers. UniGraph [25] has 180 million parameters, primarily due to its explicit integration with LLMs in encoding node features in an end-to-end way. Our GFT consistently maintains 7 million parameters during both pre-training and fine-tuning phases, making it comparable to Prodigy but significantly fewer than OFA and UniGraph.

### C.8 Detailed Illustration of Computation Tree Reconstruction

See Figure 9.

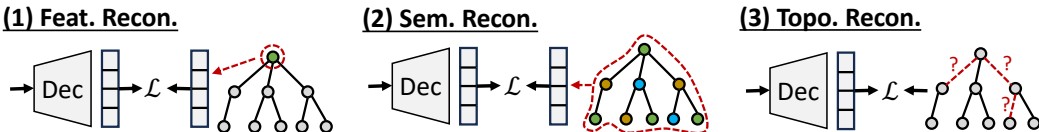

Figure 9: The detailed illustration of tree reconstruction tasks at three levels.

## D Proofs

### D.1 Proof for Theorem 2.2

We restate Theorem 2.2 from the main paper as below.

**Theorem D.1** (Transferability of Computation Tree). *Given two $L$-layer computation trees $\mathcal{T}_{v_1}, \mathcal{T}_{v_2}$ derived from the graph $\mathcal{G}$ and a GNN encoder $\phi$ with parameters $\mathbf{W} = (\mathbf{W}_1, \mathbf{W}_2)$, the Euclidean distance between the tree embeddings $\Delta \triangleq \|\phi(\mathcal{T}_{v_1}) - \phi(\mathcal{T}_{v_2})\|_2$ is bounded as follows:*

$$\Delta \leq \mathcal{C}_1 \|\mathbf{x}_{v_1} - \mathbf{x}_{v_2}\|_2 + \mathcal{C}_2 \sum_{j \in \mathcal{N}(v)} \Delta_{v_1,v_2,j}^{L-1} \leq 2\mathcal{B}_{\mathbf{x}}(\mathcal{C}_1 + \sum_{l=1}^{L} \mathcal{C}_2^l D_l) \leq 2\mathcal{B}_{\mathbf{x}} \frac{\mathcal{C}_1 - (\mathcal{C}_2 d)^L}{1 - \mathcal{C}_2 d}. \quad (9)$$

*where $\Delta_{v_1,v_2,j}^{L-1}$ represents the distance between the $L-1$-layer subtrees of the $j$-th children of nodes $v_1$ and $v_2$, and constants $\mathcal{C}_1 = \mathcal{C}_\sigma \mathcal{B}_{\mathbf{W}_1}$ and $\mathcal{C}_2 = \mathcal{C}_\sigma \mathcal{C}_\rho \mathcal{C}_g \mathcal{B}_{\mathbf{W}_2}$. Here $\mathcal{C}_\sigma, \mathcal{C}_\rho, \mathcal{C}_g$ are Lipschitz terms for GNN components, and $\mathcal{B}_{\mathbf{W}_1}, \mathcal{B}_{\mathbf{W}_2}, \mathcal{B}_{\mathbf{x}}$ denote bounded norms of $\mathbf{W}_1, \mathbf{W}_2, \mathbf{x}$, respectively. The variable $d_l$ indicates the number of children in the $l$-layer subtrees, with each $d_l \leq d$, and $D_l = d_l d_{l-1}...d_1$.*

*Proof.* We calculate the embedding distance between two $L$-layer computation trees generated from a single GNN encoder $\phi$ with parameters $\mathbf{W} = (\mathbf{W}_1, \mathbf{W}_2)$. Here we use a GraphSAGE-like encoder, as described in the Appendix F.1, that $\mathbf{W}_1$ transforms the target node, while $\mathbf{W}_2$ transforms the neighboring nodes. For simplicity, we assume that all GNN layers share the same parameters. Without loss of generality, this assumption does not affect the validity of our proofs. The term $\mathbf{x}_v$ represents the features of node $v$, and $\mathcal{N}(v)$ denotes the set of direct neighborhood in the graph, which correspond to the children of node $v$ in the computation tree $\mathcal{T}_v$.

With a bit of notation abuse, we define the GNN as:

$$\mathbf{z}_v = \phi(\mathcal{T}_v) = \sigma\Big(\mathbf{W}_1\mathbf{x}_v + \mathbf{W}_2\rho\Big(\sum_{j \in \mathcal{N}(v)} g(\mathcal{T}_j^{L-1}(\mathbf{W}))\Big)\Big) \tag{10}$$

where $\sigma$ as the non-linear activation function, $\rho$ as the permutation-invariant aggregator function, and $g$ as the update function ($\rho$ and $g$ are all based on neural networks). To simplify notation, we denote the computation tree embeddings by $\mathcal{T}(\mathbf{W}) = \phi(\mathcal{T})$. Since these functions and neural networks exhibit Lipschitz continuity, we represent their Lipschitz constants as $\mathcal{C}_\sigma$, $\mathcal{C}_\rho$, and $\mathcal{C}_g$, respectively. Additionally, we assume that the norm of node features is bounded by $\|\mathbf{W}_1\| \le \mathcal{B}_{\mathbf{W}_1}$, and the norms of model weights by $\|\mathbf{W}_1\| \le \mathcal{B}_{\mathbf{W}_1}$ and $\|\mathbf{W}_2\| \le \mathcal{B}_{\mathbf{W}_2}$.

Given the absence of constraints on the tree structures, we manually align the structures of the two trees by incorporating non-sense nodes and edges, as depicted in Figure 10. Initially, the structures of tree 1 and tree 2 are entirely distinct, as illustrated by solid lines. By integrating non-sense branches, we ensure both trees have the same structure, with three branches per node in the first layer and two in the second. These non-sense branches, considered as virtual branches, are purely for theoretical analysis convenience and hold no inherent meaning, similar to the approach in [12]. Consequently, we assume the node features of each non-sense node to be a zero vector. This alignment of tree structures enhances the coherence of subsequent analyses.

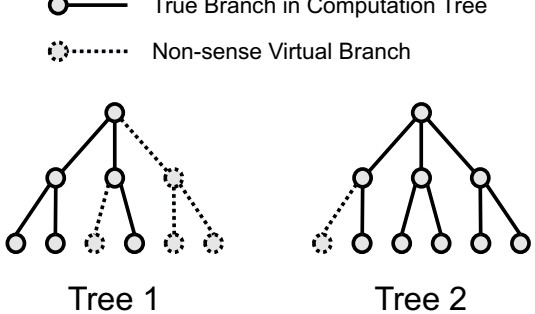

Figure 10: Adding non-sense branches to computation trees to align their structures.

Following, we expand the stability term $\Delta$:

$$\Delta \triangleq \left\| \mathcal{T}_{v_1}^L(\mathbf{W}) - \mathcal{T}_{v_2}^L(\mathbf{W}) \right\|_2$$

$$= \left\| \sigma(\mathbf{W}_1\mathbf{x}_{v_1} + \mathbf{W}_2\rho(\sum_{i \in \mathcal{N}(v_1)} g(\mathcal{T}_i^{L-1}(\mathbf{W})))) - \sigma(\mathbf{W}_1\mathbf{x}_{v_2} + \mathbf{W}_2\rho(\sum_{j \in \mathcal{N}(v_2)} g(\mathcal{T}_j^{L-1}(\mathbf{W})))) \right\|_2$$

$$\le \mathcal{C}_\sigma \left\| \mathbf{W}_1\mathbf{x}_{v_1} + \mathbf{W}_2\rho(\sum_{i \in \mathcal{N}(v_1)} g(\mathcal{T}_i^{L-1}(\mathbf{W}))) - \mathbf{W}_1\mathbf{x}_{v_2} - \mathbf{W}_2\rho(\sum_{j \in \mathcal{N}(v_2)} g(\mathcal{T}_j^{L-1}(\mathbf{W}))) \right\|_2$$

$$\le \mathcal{C}_\sigma \left\| \mathbf{W}_1\mathbf{x}_{v_1} - \mathbf{W}_1\mathbf{x}_{v_2} \right\| + \mathcal{C}_\sigma \left\| \mathbf{W}_2\rho(\sum_{i \in \mathcal{N}(v_1)} g(\mathcal{T}_i^{L-1}(\mathbf{W}))) - \mathbf{W}_2\rho(\sum_{j \in \mathcal{N}(v_2)} g(\mathcal{T}_j^{L-1}(\mathbf{W}))) \right\|_2$$

$$\le \mathcal{C}_\sigma \mathcal{B}_{\mathbf{W}_1} \left\| \mathbf{x}_{v_1} - \mathbf{x}_{v_2} \right\|_2 + \mathcal{C}_\sigma \mathcal{B}_{\mathbf{W}_2} \left\| R(\mathbf{W}, \mathcal{T}_{v_1}^L) - R(\mathbf{W}, \mathcal{T}_{v_2}^L) \right\|_2, \tag{11}$$

where $R(\mathbf{W}, \mathcal{T}_v^L) = \rho(\sum_{j \in \mathcal{N}(v)} g(\mathcal{T}_j^{L-1}(\mathbf{W})))$. We can further bound the term as:

$$\left\| R(\mathbf{W}, \mathcal{T}_{v_1}^L) - R(\mathbf{W}, \mathcal{T}_{v_2}^L) \right\|_2 \leq \mathcal{C}_\rho \left\| \sum_{i \in \mathcal{N}(v_1)} g(\mathcal{T}_i^{L-1}(\mathbf{W})) - \sum_{j \in \mathcal{N}(v_2)} g(\mathcal{T}_j^{L-1}(\mathbf{W})) \right\|_2 \quad (12)$$

As we already align the structures of two computation trees by adding non-sense branches to ensure $|\mathcal{N}(v)| = |\mathcal{N}(v_1)| = |\mathcal{N}(v_2)|$, we can merge the two terms in the RHS:

$$\left\| R(\mathbf{W}, \mathcal{T}_{v_1}^L) - R(\mathbf{W}, \mathcal{T}_{v_2}^L) \right\|_2 \leq \mathcal{C}_\rho \left\| \sum_{j \in \mathcal{N}(v)} g(\mathcal{T}_{v_1,j}^{L-1}(\mathbf{W})) - \sum_{j \in \mathcal{N}(v)} g(\mathcal{T}_{v_2,j}^{L-1}(\mathbf{W})) \right\|_2$$

$$\leq \mathcal{C}_\rho \sum_{j \in \mathcal{N}(v)} \left\| g(\mathcal{T}_{v_1,j}^{L-1}(\mathbf{W})) - g(\mathcal{T}_{v_2,j}^{L-1}(\mathbf{W})) \right\|_2$$

$$\leq \mathcal{C}_\rho \mathcal{C}_g \sum_{j \in \mathcal{N}(v)} \left\| \mathcal{T}_{v_1,j}^{L-1}(\mathbf{W}) - \mathcal{T}_{v_2,j}^{L-1}(\mathbf{W}) \right\|_2$$

$$\leq \mathcal{C}_\rho \mathcal{C}_g \sum_{j \in \mathcal{N}(v)} \Delta_{v_1,v_2,j}^{L-1}, \quad (13)$$

where $\Delta_{v_1,v_2,j}^{L-1} = \| \mathcal{T}_{v_1,j}^{L-1}(\mathbf{W}) - \mathcal{T}_{v_2,j}^{L-1}(\mathbf{W}) \|_2$.

We now establish a bound on the distance between two computation trees of identical structure by analyzing the node-wise differences from bottom to top. Denote the number of branches (i.e., children) at each $l$-layer as $d_l$, we simplify this bound as follows:

$$\left\| R(\mathbf{W}, \mathcal{T}_{v_1}^L) - R(\mathbf{W}, \mathcal{T}_{v_2}^L) \right\|_2 \leq \mathcal{C}_\rho \mathcal{C}_g d_{L-1} \max_{j \in \mathcal{N}(v)} \Delta_{v_1,v_2,j}^{L-1}. \quad (14)$$

This bound prioritizes the most influential children of a node to dominate all other branches. By combining Equation 11 with Equation 14, we recursively establish the bound of the distance between two computation trees:

$$\Delta \leq \mathcal{C}_\sigma \mathcal{B}_{\mathbf{W}_1} \left\| \mathbf{x}_{v_1} - \mathbf{x}_{v_2} \right\|_2 + \mathcal{C}_\sigma \mathcal{B}_{\mathbf{W}_2} \mathcal{C}_\rho \mathcal{C}_g d_{L-1} \max_{j \in \mathcal{N}(v)} \Delta_{v_1,v_2,j}^{L-1}$$

$$\leq \mathcal{C}_1 \left\| \mathbf{x}_{v_1} - \mathbf{x}_{v_2} \right\|_2 + \mathcal{C}_2 d_{L-1} \max_{j \in \mathcal{N}(v)} \Delta_{v_1,v_2,j}^{L-1}, \quad (15)$$

where $\mathcal{C}_1 = \mathcal{C}_\sigma \mathcal{B}_{\mathbf{W}_1}$ and $\mathcal{C}_2 = \mathcal{C}_\sigma \mathcal{B}_{\mathbf{W}_2} \mathcal{C}_\rho \mathcal{C}_g$.

Without loss of generality, we consider the distance between the original computation trees as the distance between the $L$-layer computation trees rooted at nodes $v_1$ and $v_2$, denoted as $\Delta = \Delta_{v_1,v_2}^L$. This allows us to recursively bound the distance. Given that all $\mathbf{x}$ are bounded by $\|\mathbf{x}\|_2 \leq \mathcal{B}_{\mathbf{x}}$, the distance between the node features $\mathbf{x}_{v_1}$ and $\mathbf{x}_{v_2}$ satisfies $\|\mathbf{x}_{v_1} - \mathbf{x}_{v_2}\|_2 \leq 2\mathcal{B}_{\mathbf{x}}$ by the triangle inequality. Consequently, we can further develop the recursion as follows:

$$\Delta \leq \mathcal{C}_1 \left\| \mathbf{x}_{v_1} - \mathbf{x}_{v_2} \right\|_2 + \mathcal{C}_2 d_{L-1} \max_{j \in \mathcal{N}(v)} \Delta_{v_1,v_2,j}^{L-1}$$

$$\leq 2\mathcal{B}_{\mathbf{x}}(\mathcal{C}_1 + \sum_{l=1}^L \mathcal{C}_2^l D_l), \quad (16)$$

where $D_l = d_l d_{l-1} ... d_1$.

Assuming that the number of branches (i.e., children) at each $l$-layer does not exceed the maximum number of branches in the tree, such that $d_1, ..., d_L \leq d$. We can further simplify the recursion by:

$$\Delta \leq \mathcal{C}_1 \left\| \mathbf{x}_{v_1} - \mathbf{x}_{v_2} \right\|_2 + \mathcal{C}_2 \sum_{j \in \mathcal{N}(v)} \Delta_{v_1,v_2,j}^{L-1}$$

$$\leq 2\mathcal{B}_{\mathbf{x}}(\mathcal{C}_1 + \sum_{l=1}^L \mathcal{C}_2^l D_l)$$

$$\leq 2\mathcal{B}_{\mathbf{x}} \frac{\mathcal{C}_1 - (\mathcal{C}_2 d)^L}{1 - \mathcal{C}_2 d}. \quad (17)$$

□

## D.2 Proof for Theorem 3.1

Before proving Theorem 3.1, it is necessary to first establish a more general version of the theorem, as detailed below:

**Theorem D.2.** *Given a set of instances $\{(\mathbf{x}_i, y_i)\}_{i=1}^n$ sampled from the distribution $\mathcal{P}$, and a margin-aware prototype classifier $f$ that predicts the class of instances via a distance measure. The risk $\mathcal{R}(f)$ can be bounded with probability $1 - \delta$:*

$$\mathcal{R}(f) \leq \hat{\mathcal{R}}(f) + \frac{20 \cdot \mathcal{C} \cdot p(p-1) \cdot \mathcal{B}^3 \cdot \sqrt{n}}{\rho \cdot n} + \sqrt{\frac{\ln(2/\delta)}{2n}}, \tag{18}$$

*where $\hat{\mathcal{R}}(f)$ is the empirical risk, $p$ denotes the number of tokens, $\mathcal{C}$ is a constant, $\mathcal{B}$ is the bounded norm of $\mathbf{x}$ and $\mathbf{p}$, and $\rho$ acts as the margin, serving as a penalty factor in evaluating the distance between computation trees and tokens.*

*Proof.* Given a set of tokens (prototypes), margin-based classification involves using instances to identify the nearest tokens and assigning the labels of these nearest tokens to the target instances. We denote the set of $p$ tokens by $\{\mathbf{p}_i\}_{i=1}^p$, each with a norm bounded by $\mathcal{B}$, and the labels associated with each token by $\{c_i\}_{i=1}^p$. Given an instance $(\mathbf{x}_i, y_i)$ sampled from a distribution $\mathcal{P}$, the classifier can be defined as follows:

$$f(\mathbf{x}) := c_j, \text{ where } j = \arg\min_j \|\mathbf{p}_j - \mathbf{x}\|_2^2, \tag{19}$$

where $\mathbf{x} \in \mathbb{R}^d$ is a random variable with a norm also bounded by $\mathcal{B}$, consistent with the norms of the tokens.

In this proof, we focus on a binary classification problem with only two classes, $\{-1, 1\}$. Consequently, the function can be represented as $f : \mathbb{R}^d \to \{-1, 1\}$. Without loss of generality, this binary classification setting can be readily extended to multi-class scenarios using one-versus-all or one-versus-one strategies [4]. We then define a class of functions as follows:

$$\mathcal{F} = \{f : \mathbb{R}^n \to \{-1, 1\}\}. \tag{20}$$

We denote $d_{c_+}$ as the distance to the nearest tokens with the same label $y_i = c_+$ and $d_{c_-}$ as the distance to the closest tokens with the different label $y_i = c_-$. If $d_{c_+}$ is less than $d_{c_-}$, the instance is correctly classified. Thus, the classification margin is defined as:

$$\mathcal{M}_f(x, y) := -d_{c_+} + d_{c_-}, \tag{21}$$

where a positive value indicates correct classification. Moreover, we introduce a penalty term to estimate the classification margin, defined as:

$$\mathcal{L}_{\mathcal{M}}(t) := \begin{cases} 1 & \text{if } t \leq 0, \\ 1 - \frac{t}{\rho} & \text{if } 0 < t \leq \rho, \\ 0 & \text{if } t > \rho, \end{cases} \tag{22}$$

where $\rho > 0$ is a pre-defined margin threshold.

For this classifier, the risk $\mathcal{R}(f)$ and the corresponding empirical risk $\hat{\mathcal{R}}(f)$ is defined as:

$$\mathcal{R}(f) := \mathcal{P}(f(x) \neq y), \tag{23}$$

$$\hat{\mathcal{R}}(f) := \frac{1}{n} \sum_{i=1}^n \mathcal{L}(\mathcal{M}(f(x), y)). \tag{24}$$

We can establish a Gaussian complexity bound by applying Theorem 7 from [4], which holds with a probability of at least $1 - \delta$. This is expressed as:

$$\mathcal{R}(f) \leq \hat{\mathcal{R}}(f) + \frac{2\mathcal{C}}{\rho} \cdot G_n(\mathcal{F}) + \sqrt{\frac{\ln(2/\delta)}{2n}}, \tag{25}$$

where $\mathcal{C}$ represents the Lipschitz constant, and $\rho$ represents the margin. This formulation allows us to explicitly incorporate the prediction margin into the complexity analysis. The term $G_n(\mathcal{F})$ denotes the Gaussian complexity defined over the function class $\mathcal{F}$, and an empirical Gaussian complexity can be estimated as:

$$\hat{\mathcal{G}}_n(\mathcal{F}) = \mathbb{E}_\sigma \left[ \sup_{f \in \mathcal{F}} \left| \frac{2}{n} \sum_{i=1}^n \sigma_i f(x_i) \right| \right], \tag{26}$$

where $\sigma = (\sigma_1, \sigma_2, ..., \sigma_n)$ are independent standard Gaussian random variables, $\sigma_i \sim \mathcal{N}(0, 1)$.

It is important to note that the Gaussian complexity of the function class $\mathcal{F}$ can be bounded by aggregating the complexities of all its sub-classes (Theorem 16 from [4]). In our model, the token classifier leverages $p$ tokens simultaneously; hence, it is logical to define sub-classes of $\mathcal{F}$ that utilize only two tokens for predictions. We define each sub-class as $\mathcal{F}_{ij}$, which specifically uses tokens $\mathbf{p}_i$ and $\mathbf{p}_j$ with differing labels $c_i \neq c_j$. Consequently, the total number of sub-classes is bounded by $p \cdot (p - 1)/2$. This allows us to simplify the complexity bound as follows:

$$\mathcal{R}(f) \leq \hat{\mathcal{R}}(f) + \frac{2\mathcal{C}}{\rho} \cdot p \cdot (p - 1) \cdot G_n(\mathcal{F}_{ij}) + \sqrt{\frac{\ln(2/\delta)}{2n}}. \tag{27}$$

To further simplify the bound, we must derive it for each $G_n(\mathcal{F}_{ij})$. It is important to note that $\mathcal{F}_{ij}$ can be regarded as a binary classification function class for $d_i - d_j$, where the weights are bounded:

$$\begin{aligned}
d_i - d_j &= \|\mathbf{x} - \mathbf{p}_i\|_2^2 - \|\mathbf{x} - \mathbf{p}_j\|_2^2 \\
&= (\|\mathbf{x}\|_2^2 + \|\mathbf{p}_i\|_2^2 - 2\mathbf{x}^T \mathbf{p}_i) - (\|\mathbf{x}\|_2^2 + \|\mathbf{p}_j\|_2^2 - 2\mathbf{x}^T \mathbf{p}_j) \\
&= 2\mathbf{x}^T(\mathbf{p}_j - \mathbf{p}_i) + \|\mathbf{p}_i\|_2^2 - \|\mathbf{p}_j\|_2^2 \\
&\leq 4\mathcal{B}^2 + \mathcal{B}^2 = 5\mathcal{B}^2.
\end{aligned} \tag{28}$$

Based on Lemma 22 in [4], which establishes that the empirical Gaussian complexity is bounded by a kernel function defined by $\mathcal{F}_{ij}$, we can simplify the empirical Gaussian complexity of each sub-class as follows:

$$\hat{G}_n(\mathcal{F}_{ij}) \leq \frac{10 \cdot \mathcal{B}^3 \cdot \sqrt{n}}{n}. \tag{29}$$

The difference between the Gaussian complexity and empirical Gaussian complexity is estimated to be $\epsilon$ with a probability of $2 \cdot \exp(\frac{-\epsilon^2 n}{8})$ (Theorem 11 from [4]). We can simplify the risk as follows:

$$\mathcal{R}(f) \leq \hat{\mathcal{R}}(f) + \frac{20 \cdot \mathcal{C} \cdot p(p - 1) \cdot \mathcal{B}^3 \cdot \sqrt{n}}{\rho \cdot n} + \sqrt{\frac{\ln(2/\delta)}{2n}}. \tag{30}$$

$\square$

We can readily extend the Theorem D.2 to Theorem 3.1 in the main paper.

*Proof.* Theorem D.2 establishes bounds on the generalization error of margin-based classifiers using Gaussian complexity. Analogously, vector quantization functions as a margin-based classifier by assigning instances to the nearest tokens in the vocabulary. Specifically, vector quantization utilizes this classifier for clustering, where each cluster center corresponds to a token. We assume each computation tree has a corresponding ground-truth cluster index based on the latent distribution, denoted as $\mathcal{P}_\mathcal{T}$, where $(\mathcal{T}, y) \sim \mathcal{P}_\mathcal{T}$. Thus, the vector quantization process employed in the main paper converts to a margin-based classification problem, consistent with Theorem D.2. Moreover, we can cancel the term $\mathcal{B}$ since the Cosine distance, used to measure the similarity between tree embeddings and tokens, ensures the bounded norm $\mathcal{B} = 1$. $\square$

# E  Detailed Analysis on Computation Tree Transferability

## E.1  Synthetic Dataset

**Experimental Setting.** We randomly sample node features from a uniform distribution with a dimension of 4 and conduct experiments 100 times using different seeds to report average performance.

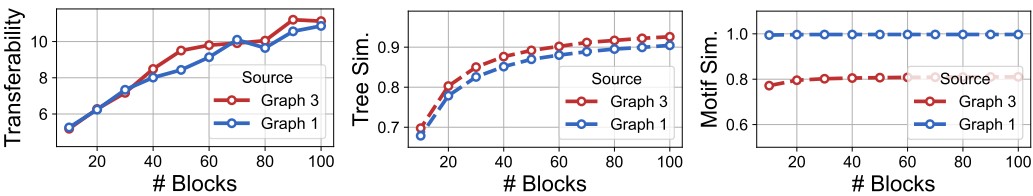

Figure 11: Transfer performance on synthetic graphs with $\mathcal{G}_2$ as the target graph.

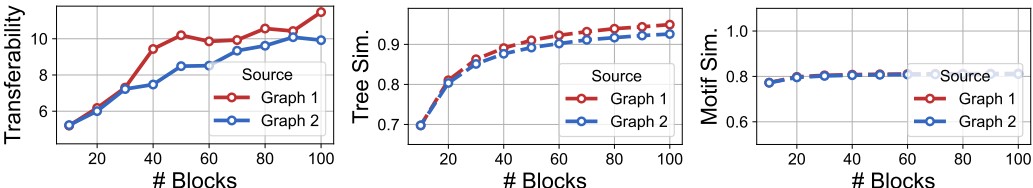

Figure 12: Transfer performance on synthetic graphs with $\mathcal{G}_3$ as the target graph.

Since the synthetic datasets do not have labels for each node, we employ a graph auto-encoder [39] for self-supervised training. The encoder is a basic 2-layer GCN model with a dimension of 4, and the decoder uses a standard inner product approach, computing the inner product between the embeddings of two nodes to determine their linkage. We set the number of training epochs at 200 and use Adam as the optimizer with a learning rate of 1e-3 and a weight decay of 0. To evaluate transferability, we use the inverse of the Central Moment Discrepancy (CMD) [102], a measure that serves as an indicator of transferability, defined as *transferability* $= \frac{1}{CMD}$. We ensure that the number of blocks in the source and target graphs is the same.

To compute the computation tree similarity (Tree Sim.), we employ the Weisfeiler-Lehman subtree kernel [64], which evaluate similarity between two graphs by considering the subtrees patterns in terms of both structure and features. To match the capabilities of the 2-layer GNN encoder used, we limit the maximum iterations of the WL subtree kernel to two. Additionally, we mitigate the impact of randomness by randomly sampling node features from a uniform distribution and repeating the process 100 times. For evaluating the motif similarity (Motif Sim.), we utilize the graphlet sampling kernel [57] with sampled graphlet size as 5.

**Additional Results.** We present additional results analyzing transferability on synthetic graphs in Figure 11 and 12. We observe that higher computation tree similarity correlates with better transferability when using $\mathcal{G}_2$ and $\mathcal{G}_3$ as target graphs. However, the impact of motif similarity is marginal. We plan to analyze link-level and graph-level tasks in future work.

### E.2    Real-world Dataset

**Experimental Setting.** We conduct transfer learning to evaluate the correlation between transferability and specific graph patterns on real-world graphs. Similar to synthetic datasets, we utilize the Weisfeiler-Lehman subtree kernel and the graphlet sampling kernel to compute tree similarity and motif similarity, respectively. We include homophily Airport graphs, consisting of `USA`, `Europe`, and `Brazil` [61], where each node represents an airport and edges denote flight connections. Nodes are labeled based on airport connectivity levels. Additionally, we also use heterophily graphs [56] that represent web links from universities such as `Cornell`, `Texas`, and `Wisconsin`, where nodes are web pages and edges are hyperlinks. The objective is to classify nodes into five categories: categories, student, project, course, staff, and faculty. In our analysis of real graphs, we consider two settings: (1) use randomly sampled node features and (2) use raw node features. This approach will offer more comprehensive insights, as node features are also related to homophily and heterophily.

The experimental settings are detailed as follows. We evaluate the transfer learning performance using a basic 2-layer GCN model with ReLU activation, running the experiments 20 times to report the average results. We pre-train the model on the source graph with 60% of nodes randomly selected and subsequently fine-tune it on the target graph with 10% of nodes randomly selected. The hidden

dimension is set to 64, and we use the AdamW optimizer with a weight decay of 1e-6. The pre-training learning rate is set at 1e-3 for all settings with 500 pre-train epochs, while the fine-tuning learning rate is set at 1e-2 for heterophily graphs and 5e-4 for homophily graphs. In the random feature setting, we sample node features from a uniform distribution across 64 dimensions. For computing graph similarity with randomly sampled node features, we conduct the experiments 100 times using different seeds. The graphlet kernel samples motifs 10,000 times, and the maximum motif size is set to 5. The maximum iteration of the subtree WL kernel is limited to 2, aligning with the number of GNN layers.

**Additional Results.**  See Table 9 for transfer learning performance on graphs with random features. Even though the node features are randomly initialized, we still observe that a high tree similarity correlates with improved transferability.

Table 9: Transfer learning performance on homophily (above) and heterophily (below) graphs with random features.

| $\mathcal{G}_{target} \rightarrow$ | Brazil | | Europe | | USA | |
|---|---|---|---|---|---|---|
| $\mathcal{G}_{source} \rightarrow$ | Europe | USA | Brazil | USA | Brazil | Europe |
| Motif Sim. | 99.01 | 92.65 | 99.00 | 96.81 | 92.68 | 96.81 |
| Acc. / Tree Sim. | 39.2 / 52.3 | 43.0 / 59.2 | 42.6 / 52.4 | 45.1 / 76.2 | 45.4 / 58.9 | 46.1 / 76.1 |

| $\mathcal{G}_{target} \rightarrow$ | Cornell | | Texas | | Wisconsin | |
|---|---|---|---|---|---|---|
| $\mathcal{G}_{source} \rightarrow$ | Texas | Wisconsin | Cornell | Wisconsin | Cornell | Texas |
| Motif Sim. | 99.97 | 99.98 | 99.99 | 99.99 | 99.98 | 99.99 |
| Acc. / Tree Sim. | 33.4 / 47.2 | 34.3 / 51.4 | 40.9 / 47.2 | 42.1 / 51.9 | 35.9 / 51.4 | 37.4 / 51.8 |

# F  Experimental Setup

## F.1  GNN Encoder

We employ a GraphSAGE-like architecture to encode node and edge features in a graph $\mathcal{G} = (\mathcal{V}, \mathcal{E})$, where node features are represented as $\mathbf{X} \in \mathbb{R}^{|\mathcal{V}| \times d_f}$ and edge features as $\mathbf{E} \in \mathbb{R}^{|\mathcal{E}| \times d_e}$. Considering a GNN with $L$ layers, the $(l+1)$-th layer node embedding for node $v$ is given by:

$$\mathbf{H}_v^{(l+1)} = \sigma \left( \mathbf{W}_1^{(l)} \mathbf{H}_v^{(l)} + \text{ReLU} \left( \sum_{u \in \mathcal{N}(v)} \mathbf{W}_2^{(l)} \left( \mathbf{H}_u^{(l)} + \varphi(\mathbf{E}_{u,v}) \right) \right) \right), \tag{31}$$

where $\mathbf{H}_v^{(l)}$ represents the node embedding at the $l$-th layer, $\mathbf{E}_{u,v}$ denotes the edge features between nodes $u$ and $v$, and $\mathbf{W}_1$ and $\mathbf{W}_2$ are the learnable matrices. The function $\varphi$, used to align feature dimensions, is chosen as the identity function, $\text{Id}(\cdot)$, in this study. While we utilize the basic GraphSAGE-like framework as the encoder, alternative, more advanced encoders such as graph attention networks [78] or other expressive GNNs [52] could potentially enhance model performance.

## F.2  Dataset

**Dataset Statistics.**  We utilize nine datasets from various domains and tasks, as detailed in Table 10. We follow the preprocessing method described in [45], using the Sentence Transformer [60] to convert raw textual descriptions of nodes and edges into 768-dimensional features. It should be noted that for knowledge graphs (KGs), we do not transform edge textual information into edge features, as the textual information already provides sufficient knowledge for KG completion.

**Dataset Splitting.**  For Cora and PubMed, we follow the common split setting with 20 labeled nodes per class for training, utilizing a predefined 10 splits with different seeds to report average performance. For WikiCS, we also employ the standard split, reporting average accuracy across 20 different training splits, each with 20 random seeds, and using 5% of nodes in each class for training. For Arxiv, HIV, and PCBA, we follow the official splits, conducting experiments 10 times

Table 10: Dataset statistics [45].

| Dataset | Domain | Task | # Graphs | Avg. #Nodes | Avg. #Edges | # Classes |
|---------|--------|------|----------|-------------|-------------|-----------|
| Cora | Citation | Node | 1 | 2,708 | 10,556 | 7 |
| PubMed | Citation | Node | 1 | 19,717 | 44,338 | 3 |
| Arxiv | Citation | Node | 1 | 169,343 | 1,166,243 | 40 |
| WikiCS | Web link | Node | 1 | 11,701 | 216,123 | 10 |
| FB15K237 | Knowledge | Link | 1 | 14,541 | 310,116 | 237 |
| WN18RR | Knowledge | Link | 1 | 40,943 | 93,003 | 11 |
| PCBA | Molecule | Graph | 437,929 | 26.0 | 28.1 | 128 |
| HIV | Molecule | Graph | 41,127 | 25.5 | 27.5 | 2 |
| ChEMBL | Molecule | Graph | 365,065 | 25.9 | 55.9 | 1,048 |

with random seeds to determine average accuracy. For WN18RR and FB15K237, we follow the splits outlined in Liu et al. [45]. Specifically, for FB15K237, the training set comprises 272,115 edges, the validation set 17,535 edges, and the test set 20,466 edges; for WN18RR, the numbers are 86,835, 3,034, and 3,134, respectively. We repeat each experiment 10 times with random seeds and report the average accuracy.

## F.3 Baseline

We compare GFT against a broad spectrum of baselines, encompassing supervised GNNs, self-supervised GNNs, graph few-shot learning, and graph foundation models.

**Supervised GNNs/MLP.** The supervised approaches include a basic MLP (Linear), GCN [40], GAT [78], and GIN [94].

**Self-supervised GNNs.** Our analysis also covers self-supervised methods for graph learning. DGI [79] utilizes contrastive learning between graph summaries and node patches. BGRL [74] employs bootstrapping to predict the same node in different views. GraphMAE [26] reconstructs node features using structural information. GIANT [11] combines language models with graph neural networks in a self-supervised fashion, achieving state-of-the-art performance.

**Graph Few-shot Learning.** To assess performance in few-shot learning scenarios, we evaluate GFT alongside methods such as GPN [15], TENT [83], GLITTER [82], and TLP [71]. Experimental results are detailed in Appendix H.

**Graph Foundation Models.** We include two primary baselines: Prodigy [30], which specializes in pre-training for in-context learning, although it is not applicable in standard pre-training and fine-tuning scenarios. For this model, we pre-train on MAG240M and evaluate performance on Arxiv, and on Wiki for FB15K237. OFA [45], in contrast, utilizes language models to align the feature spaces of different graphs and introduces a prompt graph to align task spaces, trained in a supervised manner.

## F.4 Hyper-parameter Setting

**Baselines.** For the baseline methods, we follow the hyper-parameters reported in [30, 45]. If specific hyper-parameters for a task are not reported, we set the learning rate to 5e-3 for Cora, PubMed, WikiCS, WN18RR, and HIV, and to 1e-4 for Arxiv, FB15K237, and PCBA. We configure all GNN encoders with two layers, a hidden dimension of 768, and incorporate batch normalization and ReLU activation. AdamW is used as the optimizer with a weight decay of 1e-5. For methods that utilize attention mechanisms, we specify four attention heads.

**Pre-training of GFT.** We configure our model with two layers, each having a dimension of 768, and use ReLU activation complemented by batch normalization. In vector quantization, we set the number of tokens to 128 with each token dimension at 768. We empirically determine the weights for different losses as $\beta_1 = 10$, $\beta_2 = 100$, $\beta_3 = 1$, and $\beta_4 = 0.01$. Additionally, we set the weight for

the orthogonal regularizer, $\lambda$, to 1. AdamW is utilized as the optimizer with a learning rate of 1e-4 and a weight decay of 1e-5. The pre-training phase lasts for 25 epochs with a batch size of 1024. For data augmentation, we implement an edge drop rate and a node feature drop rate, both set at 0.2. For topology reconstruction, we selectively reconstruct 10% of links and choose an equivalent number of negative samples. The sampling factor $\gamma$ for semantic reconstruction is fixed at 1.

**Fine-tuning of GFT.** We detail the hyper-parameters for different datasets in Table 11. $\lambda_{proto}$ and $\lambda_{lin}$ represent the weights of the losses for the prototype classifier and the linear classifier, respectively.

Table 11: Hyper-parameters in fine-tuning.

| Hyper-parameters | Cora | PubMed | Arxiv | Wikics | WN18RR | FB15K237 | HIV | PCBA |
|---|---|---|---|---|---|---|---|---|
| Learning Rate | 5e-4 | 5e-3 | 5e-4 | 1e-4 | 1e-3 | 5e-4 | 3e-4 | 1e-3 |
| # Epochs | 1,000 | 1,000 | 1,000 | 2,000 | 1,000 | 3,000 | 100 | 50 |
| Early Stop | 200 | 200 | 200 | 500 | 200 | 200 | 20 | 10 |
| Batch Size | 0 | 0 | 0 | 0 | 0 | 1,024 | 1,024 | 1,024 |
| # Instances per Class in $\mathbb{S}$ | n/a | n/a | n/a | n/a | n/a | 50 | 1,500 | 20 |
| $\tau$ in $f_{proto}$ | | | | 1 used for all datasets | | | | |
| $\tau$ in $f_{lin}$ | | | | 1 used for all datasets | | | | |
| $\lambda_{proto}$ | 1 | 0.1 | 1 | 1 | 0.1 | 0.1 | 0.1 | 1 |
| $\lambda_{lin}$ | 0.1 | 1 | 0.1 | 1 | 1 | 0.1 | 1 | 1 |

## F.5 Running environment

We utilize an NVIDIA A40 with 48GB GPU memory for all experiments. Both the pre-training and fine-tuning phases can be conducted on a single Nvidia GeForce RTX 3090 with 24GB memory.

## G Pre-training and Fine-tuning Results with *std*.

We report the average results of the pre-training and fine-tuning settings in the main paper. The model results along with the standard deviations are presented in Table 12.

Table 12: Model performance in pre-training and fine-tuning setting with *std*.

| Method | Node Classification | | | | Link Classification | | Graph Classification | | |
| | Cora | PubMed | Wiki-CS | Arxiv | WN18RR | FB15K237 | HIV | PCBA | *Avg.* |
|---|---|---|---|---|---|---|---|---|---|
| Linear | 58.03 ±2.33 | 68.66 ±2.24 | 70.36 ±0.58 | 66.50 ±0.14 | 78.50 ±0.59 | 87.39 ±0.07 | 66.37 ±1.11 | 72.30 ±0.34 | 71.01 |
| GCN [40] | 75.65 ±1.37 | 75.61 ±2.10 | 75.28 ±1.34 | 71.40 ±0.08 | 73.79 ±0.39 | 82.22 ±0.28 | 64.84 ±4.78 | 71.32 ±0.49 | 73.76 |
| GAT [78] | 76.24 ±1.62 | 74.86 ±1.87 | 76.78 ±0.78 | 70.87 ±0.24 | 80.16 ±0.27 | 88.93 ±0.15 | 65.54 ±6.93 | 70.12 ±0.89 | 75.44 |
| GIN [94] | 73.59 ±2.10 | 69.51 ±6.87 | 49.77 ±4.72 | 65.05 ±0.50 | 74.02 ±0.55 | 83.21 ±0.53 | 66.86 ±3.48 | 72.69 ±0.22 | 69.34 |
| DGI [79] | 72.10 ±0.34 | 73.13 ±0.64 | 75.32 ±0.95 | 69.15 ±0.20 | 75.75 ±0.59 | 81.34 ±0.15 | 59.62 ±1.21 | 63.31 ±0.89 | 71.22 |
| BGRL [74] | 71.20 ±0.30 | 75.29 ±1.33 | 76.53 ±0.69 | 71.19 ±0.18 | 75.44 ±0.30 | 80.66 ±0.29 | 63.95 ±1.06 | 67.09 ±1.00 | 72.67 |
| GraphMAE [26] | 73.10 ±0.40 | 74.32 ±0.33 | 77.61 ±0.39 | 70.90 ±0.31 | 78.99 ±0.48 | 85.30 ±0.16 | 61.04 ±0.55 | 63.30 ±0.78 | 73.07 |
| GIANT [11] | 75.13 ±0.49 | 72.31 ±0.53 | 76.56 ±0.88 | 70.10 ±0.32 | 84.36 ±0.30 | 87.45 ±0.54 | 65.44 ±1.39 | 61.49 ±0.99 | 74.11 |
| GFT | **78.62 ±1.21** | **77.19 ±1.99** | **79.39 ±0.42** | 71.93 ±0.12 | **91.91 ±0.34** | **89.72 ±0.20** | **72.67 ±1.38** | **77.90 ±0.64** | **79.92** |

## H Additional Few-shot Learning Results

We present the extended few-shot learning performance across multiple tables: Table 13, 14, 15, 16, 17, 18, and 19. In each run, we sample 20 few-shot tasks to mitigate the impact of randomness. The baselines consist of graph foundation models such as Prodigy [30] and OFA [45], alongside few-shot learning methods including GPN [15], TENT [83], GLITTER [82], and TLP [71]. In terms of graph foundation models, we compare GFT to OFA across all datasets and to Prodigy on the Arxiv and FB15K237 datasets only, as Prodigy's application is limited to these datasets by its in-context training strategy. GFT not only significantly enhances performance over Prodigy and OFA but also surpasses a broad range of specialized few-shot learning methods. Furthermore, as the number of fine-tuning instances per class increases, there is a marked improvement in model performance, demonstrating

significant adaptability to target tasks. Notably, even with extremely limited training instances, the model substantially outperforms the baselines, showcasing rapid adaptation capabilities.

Table 13: The few-shot learning performance on `Arxiv` (Part 1). We use the **bold** to indicate the performance of best baselines and best our methods. The term "# trains" indicates the number of fine-tuning instances per class.

| Method | 5-way | | | 3-way | | |
|---|---|---|---|---|---|---|
| | 5-shot | 3-shot | 1-shot | 5-shot | 3-shot | 1-shot |
| GPN [15] | 50.53 ±3.07 | 48.32 ±3.80 | 38.58 ±1.61 | 62.25 ±4.94 | 58.52 ±3.00 | 48.45 ±5.60 |
| TENT [83] | 60.83 ±7.45 | 56.03 ±8.90 | 45.62 ±10.70 | 74.20 ±9.93 | 70.48 ±11.50 | 59.38 ±13.55 |
| GLITTER [82] | 56.00 ±4.40 | 57.44 ±4.90 | 47.12 ±2.73 | 62.13 ±10.85 | 60.93 ±12.12 | 59.20 ±5.48 |
| TLP-BGRL [71] | 50.13 ±8.78 | 46.21 ±7.92 | 35.81 ±8.58 | 62.93 ±11.74 | 58.37 ±11.34 | 46.30 ±10.83 |
| TLP-SURGL [71] | **77.89 ±6.46** | **74.19 ±7.55** | **61.75 ±10.07** | **86.27 ±7.54** | **83.75 ±8.86** | **73.46 ±12.68** |
| Prodigy [30] | 61.09 ±5.85 | 58.64 ±5.84 | 48.23 ±6.18 | 73.64 ±6.93 | 71.43 ±7.28 | 61.59 ±8.53 |
| OFA [45] | 59.92 ±1.32 | 58.68 ±6.40 | 52.80 ±3.94 | 72.18 ±3.33 | 71.80 ±1.59 | 60.47 ±2.65 |
| GFT (# train = 5) | 68.00 ±1.89 | 66.00 ±2.53 | 58.20 ±4.15 | 78.56 ±4.02 | 74.00 ±3.19 | 66.22 ±4.12 |
| GFT (# train = 10) | 72.40 ±3.60 | 71.73 ±2.86 | 62.40 ±2.64 | 78.78 ±1.19 | 76.22 ±4.21 | 69.89 ±3.76 |
| GFT (# train = 20) | 73.13 ±2.80 | 71.67 ±2.43 | 64.20 ±2.07 | **80.67 ±2.34** | **79.33 ±2.34** | 72.89 ±3.01 |
| GFT (# train = 30) | **74.67 ±2.96** | **73.27 ±3.68** | **65.13 ±3.76** | 79.56 ±2.59 | 76.78 ±2.69 | **73.22 ±3.40** |

Table 14: The few-shot learning performance on `Arxiv` (Part 2).

| Method | 40-way | | | 20-way | | | 10-way | | |
|---|---|---|---|---|---|---|---|---|---|
| | 5-shot | 3-shot | 1-shot | 5-shot | 3-shot | 1-shot | 5-shot | 3-shot | 1-shot |
| Prodigy [30] | **25.51 ±0.14** | **23.69 ±0.07** | **21.44 ±0.22** | 34.29 ±0.43 | 31.30 ±0.68 | 29.21 ±0.99 | **50.84 ±1.79** | 47.39 ±2.99 | **41.05 ±6.30** |
| OFA [45] | 24.01 ±0.58 | 22.13 ±0.89 | 21.34 ±1.30 | **36.33 ±0.48** | **32.56 ±0.19** | **29.40 ±1.21** | 49.59 ±2.67 | **48.11 ±3.69** | 39.53 ±5.39 |
| GFT (# train = 5) | 36.29 ±1.04 | 34.36 ±0.95 | 26.49 ±1.11 | 45.75 ±1.10 | 42.58 ±1.17 | 35.02 ±1.00 | 56.43 ±3.45 | 52.43 ±1.13 | 44.40 ±2.60 |
| GFT (# train = 10) | 41.83 ±0.86 | 39.10 ±1.87 | 30.82 ±0.59 | 49.65 ±1.98 | 46.85 ±1.46 | 40.95 ±1.78 | 60.20 ±1.12 | 57.57 ±1.56 | 48.07 ±2.50 |
| GFT (# train = 20) | 45.07 ±1.23 | 43.90 ±1.41 | 35.02 ±1.45 | 53.25 ±1.37 | 50.85 ±1.74 | 42.97 ±1.83 | 63.47 ±1.57 | 61.37 ±2.99 | 53.23 ±1.82 |
| GFT (# train = 30) | **46.68 ±1.11** | **44.60 ±1.04** | **35.88 ±1.24** | **53.97 ±1.41** | **51.85 ±1.17** | **43.75 ±1.87** | **64.43 ±1.25** | **62.77 ±0.88** | **54.73 ±1.55** |

Table 15: The few-shot learning performance on `Cora`.

| Method | 7-way | | | 5-way | | | 2-way | | |
|---|---|---|---|---|---|---|---|---|---|
| | 5-shot | 3-shot | 1-shot | 5-shot | 3-shot | 1-shot | 5-shot | 3-shot | 1-shot |
| GPN [15] | – | – | – | – | – | – | 63.83 ±2.86 | – | 56.09 ±2.08 |
| TENT [83] | – | – | – | – | – | – | 58.97 ±2.40 | – | 54.33 ±2.10 |
| TLP-BGRL [71] | – | – | – | – | – | – | 81.31 ±1.89 | – | 59.16 ±2.48 |
| TLP-SURGL [71] | – | – | – | – | – | – | **92.49 ±1.02** | – | **81.52 ±2.09** |
| OFA [45] | 32.10 ±1.79 | 36.03 ±2.11 | 30.38 ±2.39 | 42.28 ±2.35 | 31.28 ±2.63 | 23.68 ±1.67 | 72.20 ±3.82 | **62.22 ±1.17** | 51.85 ±4.35 |
| GFT (# train = 1) | 43.55 ±7.43 | 43.31 ±8.11 | 41.40 ±8.04 | 52.30 ±6.57 | 51.47 ±6.33 | 49.80 ±6.79 | 75.00 ±4.08 | 76.33 ±3.56 | 72.92 ±4.64 |
| GFT (# train = 2) | 56.48 ±3.54 | 55.90 ±3.49 | 53.64 ±4.52 | 63.67 ±3.44 | 62.40 ±4.36 | 60.47 ±4.42 | 82.25 ±4.01 | 81.67 ±3.75 | 78.00 ±6.29 |
| GFT (# train = 5) | 67.36 ±4.31 | 67.29 ±4.39 | 66.10 ±4.39 | 74.10 ±4.26 | 74.37 ±4.53 | 72.70 ±4.87 | 87.00 ±3.36 | 86.00 ±3.33 | 86.00 ±3.35 |
| GFT (# train = 10) | **74.02 ±3.90** | **74.26 ±3.70** | **72.55 ±3.80** | **78.53 ±3.02** | **78.90 ±2.62** | **76.87 ±2.19** | **87.92 ±2.89** | **88.50 ±2.38** | **88.42 ±2.90** |

Table 16: The few-shot learning performance on `FB15K237`.

| Method | 40-way | | | 10-way | | | 5-way | | |
|---|---|---|---|---|---|---|---|---|---|
| | 5-shot | 3-shot | 1-shot | 5-shot | 3-shot | 1-shot | 5-shot | 3-shot | 1-shot |
| Prodigy [30] | 62.03 ±0.59 | 59.58 ±0.22 | 54.30 ±0.69 | **84.30 ±7.80** | 79.61 ±8.28 | 66.10 ±9.89 | 88.05 ±0.68 | 88.02 ±0.48 | 87.59 ±0.84 |
| OFA [45] | **66.51 ±0.31** | **65.76 ±0.54** | **63.48 ±0.90** | 83.64 ±6.23 | **83.14 ±1.54** | **83.46 ±4.12** | **91.43 ±0.55** | **91.12 ±0.73** | **91.01 ±0.95** |
| GFT (# train = 10) | 61.12 ±1.64 | 61.48 ±1.32 | 60.79 ±1.41 | 78.83 ±1.80 | 79.13 ±1.57 | 79.17 ±1.76 | 86.27 ±1.10 | 86.00 ±1.84 | 87.67 ±0.89 |
| GFT (# train = 20) | 70.36 ±1.73 | 70.56 ±2.12 | 70.19 ±1.44 | 85.40 ±2.10 | 85.57 ±1.29 | 85.93 ±1.48 | 91.80 ±1.07 | 91.80 ±0.62 | 91.80 ±1.54 |
| GFT (# train = 30) | **75.01 ±1.03** | **74.56 ±0.65** | **74.97 ±0.91** | **89.13 ±1.68** | **88.53 ±2.23** | **88.07 ±1.39** | **91.93 ±1.24** | **92.27 ±1.93** | **92.40 ±1.29** |

# I  Complete Ablation Study

## I.1  Pre-training Datasets

The complete results detailing the impact of different pre-training datasets are presented in Table 20, which serves as a complement to Table 5. We explore three variants: (1) pre-training on all

Table 17: The few-shot learning performance on `WN18RR`.

| Method | 10-way | | | 5-way | | | 3-way | | |
|---|---|---|---|---|---|---|---|---|---|
| | 5-shot | 3-shot | 1-shot | 5-shot | 3-shot | 1-shot | 5-shot | 3-shot | 1-shot |
| OFA [45] | 32.64 ±1.56 | 30.56 ±1.02 | 25.82 ±1.07 | 48.32 ±3.19 | 45.04 ±2.39 | 34.40 ±1.47 | 60.72 ±3.82 | 61.29 ±2.56 | 51.77 ±2.65 |
| GFT (# train = 1) | 35.50 ±4.59 | 35.50 ±5.02 | 35.33 ±4.20 | 48.80 ±3.61 | 48.53 ±3.68 | 48.13 ±4.37 | 62.56 ±2.71 | 60.67 ±3.93 | 58.44 ±3.84 |
| GFT (# train = 2) | 42.43 ±3.07 | 42.50 ±2.88 | 42.00 ±3.00 | 55.87 ±2.63 | 54.80 ±2.32 | 54.40 ±2.22 | 66.33 ±1.91 | 66.44 ±1.67 | 64.89 ±3.32 |
| GFT (# train = 5) | 44.83 ±2.88 | 44.90 ±3.07 | 44.77 ±3.50 | 58.00 ±2.61 | 57.73 ±2.18 | 57.40 ±2.53 | 68.89 ±2.19 | 69.67 ±2.10 | 68.78 ±0.96 |
| GFT (# train = 10) | 51.20 ±4.57 | 51.30 ±4.84 | 50.87 ±4.15 | 62.67 ±4.90 | 63.00 ±5.89 | 63.60 ±6.33 | 72.22 ±3.70 | 72.56 ±4.85 | 72.56 ±4.75 |

Table 18: The few-shot learning performance on `HIV`.

| Method | 2-way | | | |
|---|---|---|---|---|
| | 10-shot | 5-shot | 3-shot | 1-shot |
| OFA [45] | **54.36** ±4.90 | **57.56** ±3.66 | **59.30** ±3.04 | **57.17** ±1.82 |
| GFT (# train = 10) | 53.22 ±11.79 | 54.17 ±10.64 | 57.61 ±10.53 | 58.28 ±9.14 |
| GFT (# train = 20) | **58.67** ±7.54 | **58.78** ±6.92 | **58.44** ±7.28 | **59.94** ±7.09 |
| GFT (# train = 30) | 58.11 ±5.27 | 58.56 ±5.14 | 58.33 ±5.42 | 59.11 ±5.16 |

Table 19: The few-shot learning performance on `PCBA`.

| Method | 2-way | | | |
|---|---|---|---|---|
| | 10-shot | 5-shot | 3-shot | 1-shot |
| OFA [45] | **54.58** ±2.90 | **54.80** ±3.75 | **54.67** ±4.35 | **54.92** ±4.38 |
| GFT (# train = 10) | 55.98 ±1.39 | 56.13 ±1.32 | **56.29** ±1.31 | **56.39** ±1.49 |
| GFT (# train = 20) | **59.34** ±6.99 | **59.34** ±6.77 | 55.72 ±5.92 | 55.88 ±5.64 |
| GFT (# train = 30) | 54.81 ±2.23 | 55.14 ±2.22 | 55.18 ±2.15 | 55.33 ±1.90 |

datasets, (2) pre-training solely on the target dataset, and (3) pre-training on the remaining datasets. Our observations suggest that using only the target graph can still achieve desirable performance, as it provides graph-specific information without spurious noise. More importantly, performance improves significantly when the target graph is excluded and the remaining datasets are utilized. We hypothesize that observing more computation trees generally enhances model performance. Even without the target graph, the presence of numerous computation trees shared across various domains provides sufficient information. Moreover, using all datasets typically yields the best performance, as it offers a more comprehensive approximation of the computation tree distribution.

Table 20: Complete results of the ablation study on pre-training datasets.

| Variant | Node Classification | | | | Link Classification | | Graph Classification | | *Avg.* |
|---|---|---|---|---|---|---|---|---|---|
| | Cora | PubMed | Wiki-CS | Arxiv | WN18RR | FB15K237 | HIV | PCBA | |
| All Datasets | **78.62** ±1.21 | **77.19** ±1.99 | **79.39** ±0.42 | **71.93** ±0.12 | **91.91** ±0.34 | **89.72** ±0.20 | **72.67** ±1.38 | **77.90** ±0.64 | **79.92** |
| Target Dataset | 78.05 ±1.44 | 76.22 ±1.06 | 78.67 ±2.02 | 71.54 ±0.50 | 91.67 ±0.32 | 89.67 ±0.23 | 71.05 ±2.61 | 77.10 ±0.36 | 79.25 |
| Remaining Datasets | 77.60 ±1.07 | 75.73 ±1.63 | 78.94 ±0.89 | 71.47 ±0.22 | 91.72 ±0.18 | 89.70 ±0.22 | 72.28 ±1.83 | 77.44 ±0.27 | 79.36 |

## I.2 Pre-training Tasks

Table 21 presents complete results of model performance across different pre-training tasks, serving as a complement to Table 4. The observations align fully with those in Table 4, demonstrating that all reconstruction tasks enhance model performance compared to models without pre-training. Optimal performance is achieved when three tasks are jointly optimized.

## I.3 Strategies for Enhancing Tree Vocabulary

Table 22 presents the ablation study on strategies for enhancing the quality of tree vocabulary, as described in Section 3.1. As previously stated, both the comprehension and expressiveness of the tree vocabulary are critical properties for its effectiveness, achieved through augmentation (aug.) and an

Table 21: Complete results of the ablation study on pre-training tasks.

| Variant | Node Classification | | | | Link Classification | | Graph Classification | | *Avg.* |
|---|---|---|---|---|---|---|---|---|---|
| | Cora | PubMed | Wiki-CS | Arxiv | WN18RR | FB15K237 | HIV | PCBA | |
| n/a | 77.89 ±2.04 | 76.11 ±1.37 | 70.98 ±4.15 | 65.11 ±1.41 | 91.13 ±0.32 | 3.46 ±2.95 | 71.67 ±2.71 | 75.99 ±0.56 | 66.54 |
| w. $\mathcal{L}_{sem}$ | 78.20 ±1.72 | 75.96 ±1.53 | 79.16 ±0.91 | 71.68 ±0.16 | 91.47 ±0.31 | 89.31 ±0.22 | 72.63 ±1.81 | 77.35 ±0.10 | 79.47 |
| w. $\mathcal{L}_{feat}$ | 77.10 ±1.68 | 75.85 ±1.49 | 79.29 ±0.58 | 71.16 ±0.32 | 91.44 ±0.52 | 89.39 ±0.15 | 71.55 ±1.82 | 77.29 ±0.18 | 79.13 |
| w. $\mathcal{L}_{topo}$ | 76.42 ±1.49 | 75.82 ±1.24 | 77.95 ±0.63 | 71.81 ±0.34 | 91.08 ±0.54 | 89.48 ±0.09 | 72.02 ±1.62 | 77.11 ±0.26 | 78.96 |
| All Tasks | **78.62** ±1.21 | **77.19** ±1.99 | **79.39** ±0.42 | **71.93** ±0.12 | **91.91** ±0.34 | **89.72** ±0.20 | **72.67** ±1.38 | **77.90** ±0.64 | **79.92** |

orthogonal regularizer (ortho. reg.), respectively. We note that removing any component results in a degradation in model performance.

Table 22: Complete results of the ablation study on strategies for enhancing tree vocabulary.

| Variant | Node Classification | | | | Link Classification | | Graph Classification | | *Avg.* |
|---|---|---|---|---|---|---|---|---|---|
| | Cora | PubMed | Wiki-CS | Arxiv | WN18RR | FB15K237 | HIV | PCBA | |
| All Components | **78.62** ±1.21 | **77.19** ±1.99 | **79.39** ±0.42 | **71.93** ±0.12 | **91.91** ±0.34 | **89.72** ±0.20 | **72.67** ±1.38 | **77.90** ±0.64 | **79.92** |
| w/o Aug. | 77.44 ±1.35 | 76.00 ±1.80 | 78.76 ±0.73 | 71.50 ±0.34 | 91.11 ±0.38 | 89.07 ±0.21 | 71.54 ±2.89 | 77.50 ±0.15 | 79.12 |
| w/o Ortho. Reg. | 77.64 ±1.91 | 76.94 ±2.01 | 78.81 ±0.78 | 71.44 ±0.43 | 91.23 ±0.43 | 89.29 ±0.22 | 70.84 ±2.91 | 77.04 ±0.31 | 79.15 |

## I.4 Fine-tuning Tasks

We analyze the impact of different computation tree classification tasks for fine-tuning. The complete results are presented in Table 23, which complements Table 4. Specifically, GFT employs both a linear classifier and a prototype classifier to utilize information from various levels of the tree. The prototype classifier excels in node-level tasks, while the linear classifier performs better in the other two tasks. However, combining these two methods yields the best performance.

Table 23: Complete results of the ablation study on fine-tuning tasks.

| Variant | Node Classification | | | | Link Classification | | Graph Classification | | *Avg.* |
|---|---|---|---|---|---|---|---|---|---|
| | Cora | PubMed | Wiki-CS | Arxiv | WN18RR | FB15K237 | HIV | PCBA | |
| w. $f_{proto}$ | 78.53 ±1.21 | 77.13 ±1.40 | 79.36 ±0.54 | 71.53 ±0.41 | 76.76 ±1.01 | 0.65 ±0.00 | 56.21 ±4.24 | 63.56 ±1.06 | 62.97 |
| w. $f_{lin}$ | 76.52 ±3.50 | 76.82 ±1.96 | 78.35 ±1.07 | 70.36 ±0.57 | 90.05 ±2.34 | 87.93 ±1.60 | 69.06 ±4.98 | 75.36 ±2.86 | 78.06 |
| All Tasks | **78.62** ±1.21 | **77.19** ±1.99 | **79.39** ±0.42 | **71.93** ±0.12 | **91.91** ±0.34 | **89.72** ±0.20 | **72.67** ±1.38 | **77.90** ±0.64 | **79.92** |

Table 24: Complete results of the ablation study on tree vocabulary.

| Variant | Node Classification | | | | Link Classification | | Graph Classification | | *Avg.* |
|---|---|---|---|---|---|---|---|---|---|
| | Cora | PubMed | Wiki-CS | Arxiv | WN18RR | FB15K237 | HIV | PCBA | |
| # Tokens = 128 | 78.62 ±1.21 | 77.19 ±1.99 | 79.39 ±0.42 | 71.93 ±0.12 | **91.91** ±0.34 | 89.72 ±0.20 | **72.67** ±1.38 | 77.90 ±0.64 | 79.92 |
| # Tokens = 256 | 78.24 ±1.65 | **77.26** ±1.93 | 79.31 ±0.70 | 72.02 ±0.28 | 91.90 ±0.32 | 89.82 ±0.21 | 72.22 ±1.80 | **78.11** ±0.26 | 79.86 |
| # Tokens = 512 | **78.86** ±1.22 | 77.17 ±1.20 | **79.54** ±0.52 | **72.19** ±0.20 | 91.75 ±0.27 | **89.96** ±0.15 | 72.36 ±0.92 | 78.06 ±0.38 | **79.99** |
| w/o. Vocab. | 78.27 ±1.32 | 76.43 ±2.59 | 78.06 ±0.38 | 70.82 ±0.08 | 81.52 ±0.15 | 91.87 ±0.05 | 56.12 ±7.44 | 82.21 ±0.13 | 76.91 |

## I.5 Tree Vocabulary

Table 24 presents a detailed analysis of model performance with different numbers of tokens and without utilizing the tree vocabulary, complementing Table 6. Although increasing the number of codes can enhance performance to a certain extent, it is not necessarily effective in all scenarios. Specifically, in only four out of eight scenarios, the maximum number of codes (512 tokens) yields the best results. This observation is consistent with our theoretical analysis, suggesting that more codes may increase the upper bound of generalization error, potentially due to overfitting risks. Furthermore, this phenomenon might also be attributed to the limited diversity of datasets; the eight utilized datasets originate from only four domains (citation, web link, knowledge graphs, and molecules). For these

domains, a smaller number of codes may suffice. We hypothesize expanding the number of domains might necessitate more codes; we intend to explore in future work. Regarding the variant that does not use vocabulary, we bypass vocabulary training during pre-training, and directly append a linear classifier behind the GNN for classification during fine-tuning. The results indicate that using the vocabulary significantly enhances model performance, particularly in link- and graph-level tasks, aligning with our theoretical considerations regarding the generalizability of tree tokens.

