# OpenReview forum: "GFT: Graph Foundation Model with Transferable Tree Vocabulary"
_NeurIPS.cc/2024/Conference — NeurIPS 2024 poster_

### Official Review · Reviewer_hEws · 2024-07-02

**Soundness:** 3
**Presentation:** 3
**Contribution:** 3
**Rating:** 5
**Confidence:** 4

**Summary:**

The paper proposes GFT, a graph foundation model based on computation tree. Extensive experiments and theoretical analyses are conducted to show the effectiveness of GFT across diverse tasks.

**Strengths:**

1. Addresses a significant challenge (identifying transferrable patterns) in graph foundation models.
2. Includes both theoretical and empirical evaluations on synthetic and real-world graphs.
3. Conducts extensive experiments including multiple graph tasks and domains.

**Weaknesses:**

1. The experiments are unconvincing (and most likely unfair): Only a few outdated supervised baselines are used, and their results are weaker than expected. For instance, the GCN's performance on Cora should surpass 80, and the results for GIANT on Arxiv are notably low. Are all baselines using SentenceBERT embedding? If not and the raw features are used, the improvements might be contributed to better text encoders.
2. Assumes a common feature space across different graphs, limiting its applicability to non-textual graphs.
3. The method is complex, involving a broad hyperparameter space, including computation tree parameters and the betas in Equation 3.
4. Certain sections of the paper are unclear, such as the sampling/construction of the computation trees and the interpretation of the y-axis in Figure 5.
5. The distinction between the computation tree and subgraph is not clearly defined.
6. The analysis of time complexity should consider the size of subgraphs, similar to GraphSAGE. Furthermore, the discussion should focus on actual wall time to better demonstrate the efficiency of the GFT.

**Questions:**

How are baseline results obtained? What kind of features are leveraged?

---

> ### Author Rebuttal · Authors · 2024-08-05
>
> We appreciate the reviewer’s recognition of our work and thank the reviewer for detailed feedback. We address your concerns as follows.
>
> > W1: The experiments are unconvincing and most likely unfair.
>
> [W1.1] *Outdated supervised baselines.* The baselines are not outdated, as our primary comparisons are with graph foundation models (GFMs) such as Prodigy [1] and OFA [2], published in NeurIPS 23 and ICLR 24, respectively. For supervised and self-supervised baselines, we selected classic works including GCN, GAT, GIN, DGI, BGRL, GraphMAE, and GIANT, in line with existing GFMs [1-4]. Despite being compared to recent supervised methods, our GFT still achieves better average performance (79.92) compared to GATv2 (ICLR ’22) with 76.19 and RevGAT (NeurIPS ’22) with 74.97.
>
> [W1.2] *What features are used?* We strictly followed the OFA [2] to generate node attributes using SentenceBERT and conducted experiments on all baselines based on the SentenceBERT features rather than original features, ensuring a convincing and fair comparison. Since all baselines use the same text embeddings, we can ensure that any improvement is not merely due to the text encoder.
>
> [W1.3] *Why the performance of baselines is lower than expected?* It's becuase the text encoder does not always lead to better performance with three resaons: (1) **Complexity of text-based continuous features.** These features are more complex than manually designed (discrete) features, making it challenging to learn the underlying correlations between features and labels. References such as [6] indicate this issue, where the performance of GraphSAGE on Arxiv degrades from 71.49 to 70.97 when replacing the raw 128-dim features with 768-dim BERT embeddings. Another example is in Table 10 of [5], where performance on molecule graphs (HIV) degrades from 75.52 to 74.20 when using textual features. (2) **Noisiness of text-based features.** Text-based features can be noisier than manually designed features due to biases in the prompts fed into language models, which can impair the quality of generated features and lead to spurious correlations between features and labels. (3) **Harder experimental setting.** For example, we use standard splits with 20 labeled nodes per class for training on Cora and Pubmed, rather than the 10/10/80 train/val/test splitting.
>
> The lower performance of self-supervised baselines is due to their pretraining on all graphs, which may introduce extra noise, confusing the models in identifying patterns needed for specific tasks or domains. For example, [5] indicates that models (GraphMAE, DGI) pretraining on multiple graphs perform significantly worse than pretraining on a single graph, and [4] shows that GIANT’s performance on Arxiv is 70.87 when the model is pre-trained on ogbn-Papers100M.
>
> > W2: Assumes a common feature space across different graphs, limiting its applicability to non-textual graphs.
>
> Although we use text-attributed graphs in our experiments, our GFT can also be applied to non-textual graphs by employing techniques like [7] to align the feature space. Furthermore, this assumption does not significantly impact the contribution of our work, as our primary goal is to identify potential transferable structures across graphs.
>
> > W3: The method involves a board hyperparameter space.
>
> We thank the reviewer for mentioning the potential efforts involved in hyperparameter tuning. However, our method does not require extensive hyperparameter tuning. For example, when balancing the weights of different losses, it is sufficient to adjust their scales. For other hyperparameters, we can follow existing papers to set the default value. This simplifies our efforts to tuning only a few hyperparameters, such as learning rate, weight decay, and batch size. Despite this, our GFT still achieves superior performance.
>
> > W4: The interpretation of y-axis in Figure 5.
>
> This axis represents a metric (NT gap) used to measure the occurrence of negative transfer. The metric follows the principle: for a certain target task T, if the pretrained model (pretrained on S) performs worse than a randomly initialized model, negative transfer occurs. Therefore, we define this metric as $\text{NT gap} = R(S,T) - R(\emptyset, T)$, where $R(S,T)$ is the loss of the pretrained model and $R(\emptyset, T)$ is the loss of the initialized model for the downstream task T.
>
> > W5: The difference between computation trees and subgraphs. How to construct computation trees?
> >
> > W6: The time complexity analysis should consider the size of subgraphs. The real-time consumption should be provided.
>
> Please refer to [W1] of the general response, where we provide a detailed comparison between computation trees and subgraphs, including the construction and time consumption. We would like to further clarify why it is not necessary to consider the size of subgraphs. The reason is that matrix-based BFS has the same time complexity regardless of the size of the subgraphs. Moreover, given a node, the number of involved nodes within k-hop is the same, no matter in tree-like or subgraph-like structures. This makes the impact of the subgraph size marginal when comparing computation trees and subgraphs.
>
> Best regards,
>
> The authors
>
> ---
>
> Reference:
>
> [1] Huang, et al., PRODIGY: Enabling In-context Learning Over Graphs, NeurIPS, 2023.
>
> [2] Liu, et al., One For All: Towards Training One Graph Model for All Classification Tasks, ICLR, 2024.
>
> [3] Chen, et al., LLaGA: Large Language and Graph Assistant, ICML, 2024.
>
> [4] He, et al.,UniGraph: Learning a Cross-Domain Graph Foundation Model From Natural Language, Arxiv, 2024.
>
> [5] Chen, et al., Text-space Graph Foundation Models: Comprehensive Benchmarks and New Insights, Arxiv, 2024.
>
> [6] Chien, et al., Node Feature Extraction by Self-Supervised Multi-scale Neighborhood Prediction, ICLR, 2022.
>
> [7] Zhao, et al., All in One and One for All: A Simple yet Effective Method towards Cross-domain Graph Pretraining, KDD, 2024.

---

> ### Author Response · Authors · 2024-08-10
>
> Dear Reviewer hEws,
>
> Thank you for your comments and suggestions to improve our paper, we would like to check whether our response answers your questions. Following your comments, we comprehensively discuss the experimental settings and the results obtained, and also provide a point-to-point response to all of your questions. We look forward to your reply.

---

> ### Comment · Reviewer_hEws · 2024-08-10
> **Response to rebuttal**
>
> I have carefully reviewed the rebuttal and the attached PDF. I appreciate the authors' efforts in addressing the weaknesses I identified.
>
> The main concern, namely that the improvement is largely due to a better text encoder, has been mostly addressed. However, the results still fall short of my expectations. For comparison, the performance on GCN and GAT reported in [this paper](https://arxiv.org/pdf/2307.03393) is 82.20 ± 0.49 and 82.77 ± 0.59, respectively (see Table 1 for details).
>
> The explanation provided for the GIANT performance is unconvincing and supports my earlier assumption that the experimental setup is biased toward the GFT model. GIANT was designed to train on a single graph and should be evaluated in that context.
>
> Overall, since my primary concern has been sufficiently addressed, I have increased my score accordingly.

---

> > ### Author Response · Authors · 2024-08-11
> >
> > Thanks for your constructive comments again and for raising your score! Your review is really helpful for our work!

---

### Official Review · Reviewer_M9ZQ · 2024-07-07

**Soundness:** 3
**Presentation:** 2
**Contribution:** 3
**Rating:** 6
**Confidence:** 4

**Summary:**

This paper explores the concept of the transferable token in graph foundation models. Specifically, the paper proposes to use the computation tree as the transferable token for graph learning and prove its efficiency from both theoretical and empirical perspectives. Then, the paper proposes GFT model, GFT first pretrain a graph tokenizer using vector quantization to tokenize text-attributed graphs across different domains through a computation tree. Next, it is fine-tuned on downstream tasks. The model shows great results on generalization ability, especially on few-shot and zero-shot experiments.

**Strengths:**

1. Investigating transferable tokens in graph learning is important and fundamental to the community. This paper is the first one that demonstrates the potential existence of the transferable token in text-attributed graphs.
2. The paper is well-structured with solid theoretical and experimental results to demonstrate the advantages of the proposed model.

**Weaknesses:**

1. In Theorem 2.2, the distance bound is associated with the distance between the $j$-th neighbor of node $v_1$ and $v_2$. How to define the $j$-th neighbor related to two nodes? What if the two nodes have different numbers of neighbors?
2. In Theorem 2.2, the distance bound is related to the distance of the feature between two nodes (as well as the distance between node neighbors, as it is defined recursively). However, in the real world, the distance of the feature itself can be unbounded, which makes Theorem 2.2 only applicable when features are close and less useful in real scenarios (Even in text-attributed graphs, sentence embedding from different domains can still be too diverse).
3. For Section 3.1, I am not sure if I understand it wrongly or if I missed something. There are a few parts I think it is not true. For Equation 2, should loss compute between $z_i$ and $c_i$, instead of $q_i$? Otherwise, it does not make sense to me. Meanwhile, what is the $\delta_j$, I cannot find its definition in the paper. Please correct me if I am wrong.

4. Although the authors did extensive ablation studies, there is one I am particularly interested in about cannot find in the paper. How well can be model be if the pretraining datasets are totally from different domains of the downstream tasks? For example, what is the performance if the model is pre-trained on molecular datasets but tested on citation networks or vice versa?

**Questions:**

1. I am curious about Table 23. Since the graph tokenizer needs to tokenize both structural and textual information. The token size should be much larger than 512 even within a single domain from my sense given the abundance of both the graph structure and textural information. Could authors explain more about it? Have authors explore what are these tokens learned exactly?

2. The approach in this paper is somewhat similar to the VQGraph [1]. It is worth discussing the relation and difference between the two methods.

3. The author claims the proposed method does not need to extract subgraphs. I am wondering how is the method deal with the case like the graph is too large to fit into one GPU?


[1] Yang et al., VQGRAPH: RETHINKING GRAPH REPRESENTATION SPACE FOR BRIDGING GNNS AND MLPS, ICLR 2024.

**Limitations:**

There are no major limitations from my perspective. I encourage the authors to further explore the proposed methods, like using more expressive GNNs or pre-train on large-scale datasets, as discussed in the limitation section.

---

> ### Author Rebuttal · Authors · 2024-08-04
>
> We greatly appreciate Reviewer M9ZQ for the insightful feedback and are deeply inspired by your recognition of our contribution as “the first to demonstrate the potential existence of transferable tokens in text-attributed graphs.” We address your concerns as follows.
>
> > W1: How to define the 𝑗th neighbor related to two nodes in Theorem 2.2?
> >
>
> Let’s consider two cases. If the number of neighborhoods of the two nodes is the same, we can readily compare their corresponding neighbors. If the number of neighborhoods is different (e.g.,  n1,  n2  where  n1 > n2 ), we use a straightforward method for approximation. For the node with fewer neighborhoods (n2), we add virtual nodes in the number of (n1-n2) to ensure all nodes have n1 neighborhoods. It is true that a large gap in the number of neighborhoods between two nodes may lead to significant approximation error. However, we believe the approximation error would not be too large in real-world cases, where node degrees generally follow long-tail distributions. That is, most nodes have small degrees, while a small number of central nodes have high degrees. Therefore, it is not necessary to add too many virtual nodes when measuring the distance between computation trees, mitigating the approximation error.
>
> > W2: How to ensure the bounded distance measurement in Theorem 2.2?
> >
>
> While the distance between original node features might be unbounded, we employ a simple normalization technique (which is actually used in this paper) to manually bound the distance. Specifically, we normalize the features as $\mathbf{x} / \| \mathbf{x} \|$, converting the unbounded Euclidean distance into a bounded Cosine distance. This makes Theorem 2.2 more practical.
>
> > W3: The issue in Eq. 2 and the definition of 𝛿𝑗.
> >
>
> We apologize for the confusion and sincerely thank you for identifying the typos in the paper. You are correct that the loss is computed between  $z_i$  and  $c_i$, consistent with the vanilla VQ-VAE. We will correct these typos.
>
> Regarding $\delta(\cdot)$ , it is a linear projection implemented by a basic fully-connected layer. We will refine the paper to provide better clarification.
>
> > W4: What’s the model performance when the pre-training domains are totally different from downstream tasks?
> >
>
> We thank the reviewer for raising this important point. We conducted experiments on two citation networks, Cora and Arxiv, and two molecule networks, HIV and PCBA. The results are shown below:
>
> | Pretrain Data | Cora | Arxiv | HIV | PCBA |
> | --- | --- | --- | --- | --- |
> | Cora + Arxiv | 78.72 ± 1.62 | 71.85 ± 0.12 | 70.62 ± 3.06 | 78.10 ± 0.40 |
> | HIV + PCBA | 78.46 ± 1.42 | 71.76 ± 0.15 | 72.19 ± 2.04 | 78.43 ± 0.25 |
>
> Even when the pre-training and downstream datasets are from different domains, there is no significant model degradation. For example, the model performance on Cora is 78.72 when pre-trained on citation networks and 78.46 when pre-trained on molecule networks. Additionally, the performance on PCBA is 78.43 when pre-trained on molecule graphs and 78.10 when pre-trained on citation networks.
>
> > W5: Why the token size is smaller than expected? What are tokens learn exactly?
> >
>
> One potential reason for the relatively lower number of tokens is the use of a prototype classifier. The construction of prototypes in this classifier involves combining different tokens in a memory bank, which can be interpreted as generating new tokens from previous tokens. In other words, the prototype classifier implicitly enlarges the vocabulary and enhances its diversity. Removing the prototype classifier degrades the average model performance from 79.92 to 78.06, while increasing the token size to 2048 provides only a marginal performance gain, reaching 80.38.
>
> We also explored the meaning of tokens, as shown in Figure 1 of the attached PDF file in the general response. This figure reveals a set of tokens shared across datasets, each potentially maintaining general patterns across domains and tasks. Additionally, distinct tokens are used by individual domains or datasets, indicating domain-specific patterns. These well-separated tokens between different domains might provide a better source for the prototype classifier in creating prototypes.
>
> > W6: Comparison to VQGraph.
> >
>
> We thank the reviewer for the suggestion to compare our GFT to VQGraph and discuss their relation and differences. The major connection between GFT and VQGraph is the usage of vector quantization in learning a discrete vocabulary for downstream tasks. However, there are four major differences between GFT and VQGraph. (1) Model Objective: GFT focuses on building a general task reasoner, but VQGraph aims to train a structure-aware MLP for efficient inference. (2) Pretrain Dataset: GFT is pre-trained on cross-domain and cross-task datasets to acquire transferable patterns among graphs, but VQGraph is pre-trained on a single dataset to better capture the structural information. (3) Usage of Tokens: GFT treats tokens as specific transferable patterns, using them directly to build classifiers. VQGraph, on the other hand, treats tokens as external structural knowledge to complement the training of MLP classifiers. (4) Downstream Tasks: GFT can be applied to various graph-related tasks with different settings like few-shot and zero-shot learning. VQGraph is designed for node classification with a basic pre-training and fine-tuning setting.
>
> > W7: How GFT fits on with large graphs?
> >
>
> GFT can still use mini-batch to fit to a large graph. Given a large graph, we extract the mini-graph through mini-batch and use GNNs to encode the computation trees preserved on the mini-graph. Please refer to [W1] of general response for details.
>
> Best regards,
>
> The authors

---

> ### Author Response · Authors · 2024-08-10
>
> Dear Reviewer M9ZQ,
>
> Thank you for your comments and suggestions to improve our paper, we would like to check whether our response answers your questions. Following your comments, we provide detailed responses to your questions and present a comparison to VQGraph based on your recommendation. We look forward to your reply.

---

> > ### Comment · Reviewer_M9ZQ · 2024-08-10
> >
> > I would like to thank the authors for their detailed reply to all my concerns. Most of my concerns are clarified after rebuttal and I would like to maintain my original score. Overall, Although after the rebuttal I still think the theoretical analysis has some flaws (particularly the neighbor part in Theorem 2.2 as graphs in different domains could indeed have very different distributions even in the real world), I do believe the merits of the paper outweigh the limitations.

---

> > > ### Author Response · Authors · 2024-08-11
> > >
> > > Thanks for your constructive comments again! Your review inspires us to refine our Theorem 2.2 to have a better approximation for handling real-world graphs with different distributions in our future work!

---

### Official Review · Reviewer_U8xo · 2024-07-13

**Soundness:** 3
**Presentation:** 3
**Contribution:** 3
**Rating:** 6
**Confidence:** 3

**Summary:**

This paper proposed a novel computation tree method to improve the transferability between the pre-train model and downstream tasks. This paper rethinks the transferable pattern in graphs as computation trees and validate their transferability both empirically and theoretically. The proposed GFT leverages computation tree reconstruction to acquire general graph knowledge from cross-domain datasets and uses computation tree classification to facilitate adaptation to various target tasks Comparing with other previous methods, the proposed GFT can improve model generalization and reduce the risk of negative transfer, which is suitable for the cross-domain and cross-task situation.

**Strengths:**

a. Originality: The computation tree has been proposed in several previous works that have been adequately cited in this paper. The author reconstructs the computation tree in a new way and combines it with Vector Quantization method, effective and preventing over-fitting. b. Quality: The paper is technically sound, and the loss function proposed is comprehensive, enabling a deep understanding of the structural and semantical attribute of computation trees. The experiments in the manuscript are very comprehensive and effectively demonstrate the superiority of GFT, and important claims are well supported by theoretical analysis. c. Clarity: The manuscript is well organized and clearly written, although some descriptions could be clearer. d. Significance: The experimental results advance the SOTA methods, and the computation tree method proposed is easy for other researchers to use.

**Weaknesses:**

It is not very clear how this work differs from previous contributions. The explanation of the superiority of computation trees over subgraphs is unconvincing because the computation tree in this paper is very different from some proposed computation trees (e.g. junction tree, H-tree, etc.) but more like the subgraph. It would be better if the author could provide detailed comparison and analysis between the computation tree and the subgraph; The quality is a bit limited. The Lfeat is to minimize the discrepancy between the local structure of a node and its feature, but is there any inherent relationship between the two? Besides, the method might not be able to deal with link prediction in a good manner. In a graph, if node va and vb are isomorphic while the links (va, vc) and (vb, vc) are not isomorphic, the vanilla GNN with the same node representations va and vb gives the same prediction to links (va, vc) and (vb, vc), but GFT does not address this issue; The clarity should be improved. The explanation for Ltree should be detailed. For example, what are decoders used for q projection? In line 269, the definition of R(f) is not given; The significance of the paper is slightly limited. It might be difficult to extend the GFT to zero-shot scenarios because the fine-tune process is necessary for GFT but it will be excluded in zero-shot scenarios.

**Questions:**

1. In Appendix C.5, the paper argues that the subgraph method incurs additional time and memory costs. Why can the proposed computation tree avoid these problems?

2. The Lfeat is to minimize the discrepancy between the local structure of a node and its feature. Why do you use Lfeat and is there any intrinsic connections between the local structure of a node and its feature?

3. For Ltree, what are decoders used for q projection?

4. In a graph, if node va and vb are isomorphic while the links (va, vc) and (vb, vc) are not isomorphic, the vanilla GNN with the same node representations va and vb gives the same prediction to links (va, vc) and (vb, vc). How does GFT address this issue in link prediction?

5. What is the definition of R(f)?

6. From the experimental results, GFT outperforms other methods. Is this due to the computation tree or the more comprehensive loss function? Would the overall performance be worse if the computation trees were replaced with subgraphs? 7. Is it possible for GFT to be applicable in zero-shot scenarios?

---

> ### Author Rebuttal · Authors · 2024-08-05
>
> Dear Reviewer U8xo,
>
> We appreciate the reviewer’s recognition of originality, quality, clarity, and significance of our work, and thank the reviewer for detailed feedback. We address your concerns as follows.
>
> > W1: Differences between computation trees and subgraphs.
>
> Please refer to the [W1] in the general response.
>
> > W2: How $L_{feat}$ works?
>
> $L_{feat}$ uses localized information to predict the original node features via a linear predictor. This can potentially model the positive or negative correlations between neighborhoods and the target node, corresponding to homophily and heterophily, respectively. Let $z$ be the learned embedding and $x$ be the raw features. If the neighborhood information is similar to the target node (homophily), the predictor might learn a nearly identical function since $z$ might be similar to $x$ . However, under heterophily, where $z$ and $x$ are far from each other, the linear predictor might learn a distinct mapping function to model such correlation. Empirically, [1] proposes a pre-training task similar to $L_{feat}$ and achieves desirable performance on node classification and graph classification. To elucidate this further, we provide a thorough ablation analysis to validate the effectiveness of $L_{feat}$ in our GFT:
>
> ||Arxiv (Citation)|WN18RR (KG)|HIV (Molecule)|
> |---|:---:|:---:|:---:|
> |No Pretrain|65.11|91.13|71.67|
> |Only $L_{feat}$|71.16|91.44|71.55|
> |w.o. $L_{feat}$|70.41|91.33|72.04|
> |All Tasks|71.93|91.91|72.67|
>
> The $L_{feat}$ is effective in node-level tasks (Arxiv) where papers tend to cite related papers, following the homophily principle. It also benefits link-level tasks (KG reasoning), where two entities sharing similar types of relationships may be connected. However, for graph-level tasks that may require a global understanding of patterns and motifs, $L_{feat}$ may be less important. Despite this, removing $L_{feat}$ still degrades model performance. These experiments emphasize the benefit of modeling the correlations between the local structures of a node and its features.
>
> > W3: What are decoders used for q projection?
>
> The q projection is implemented by a basic fully-connected layer. We apologize for any confusion and sincerely thank you for identifying this issue. We will refine the paper to provide better clarification.
>
> > W4: How to handle the case that nodes are isomorphic but edges are not isomorphic.
>
> We greatly appreciate this insightful feedback. Similar to previous GFMs with basic GNN encoders [2, 3], our GFT cannot enhance model expressiveness to handle such link-level isomorphism. However, this limitation arises primarily due to the limited expressiveness of the backbone model (GraphSAGE) rather than our model design, and thus **does not significantly impair the contribution of the overall framework**. We can address this issue by using more expressive GNN backbones as described in [4]. Moreover, such conditions may not always occur in real-world applications with rich node attributes, where two nodes are less likely to have the same embeddings, thereby preventing node isomorphism.
>
> > W5: What is the definition of R(f)?
>
> The definition of R(f) is related to the VQ operation. Specifically, VQ clusters the input vectors by measuring the distance between the input vectors and token vectors in the vocabulary, where the process can be interpreted as a margin-aware prototype classifier $f$ [7]. The classifier $f$ aims to classify the inputs into their latent clusters $y$ , and the predictions are denoted as $\hat{y}$. Based on that, R(f) measures the risk of the classifier $f$. Note that this analysis is based on the extension of the latent variable assumption, a general assumption in statistics and machine learning that variables are sampled from a latent distribution, where each input is assigned a label indicating its latent cluster.
>
> > W6: What’s the impact of the comprehensive loss functions? What will happen if computation trees are replaced by subgraphs?
>
> GFT surpasses baselines even with a single loss function, and replacing computation trees with subgraphs impairs model performance. We select two basic baselines, including GAT and GraphMAE, and implement a subgraph version of GFT (GFT - Subgraph) following [6]. We further extend our comprehensive objective to the subgraph version (GFT - Subgraph + Our Task) as another baseline. The results are as follows:
>
> |  | Node | Link | Graph | Avg. |
> |---|:---:|:---:|:---:|:---:|
> | *Model Performance with linear classifier* |||||
> |GAT|74.69|84.55|67.83|75.44|
> |GraphMAE|73.98|82.15|62.17|73.07|
> |GFT - Subgraph|73.61|85.73|64.67|74.41|
> |GFT - Subgraph + Our Task|74.23|86.49|67.89|76.13|
> |GFT|**75.51**|**88.99**|**72.21**|**78.06**|
> | *Impact of Loss Objectives* |||||
> |Only $L_{sem}$|76.25|90.39|74.99|79.47|
> |Only $L_{feat}$|75.85|90.42|74.42|79.13|
> |Only $L_{topo}$|75.50|90.28|74.57|78.96|
> |GFT (All Tasks)|**76.78**|**90.82**|**75.29**|**79.92**|
>
> The results demonstrates the benefit of using computation trees even without a comprehensive loss function, referring to *Impact of Loss Functions*. When the computation trees are replaced by subgraphs, the model performance degrades even with our comprehensive tasks, further supporting the advantage of using computation trees as basic patterns.
>
> > W7: Zero-shot learning.
> >
>
> Please refer to [W2] in the general response.
>
> Best regards,
>
> The authors
>
> ---
>
> Reference:
>
> [1] Hou, et al., GraphMAE: Self-Supervised Masked Graph Autoencoders, KDD, 2022.
>
> [2] Liu, et al., One For All: Towards Training One Graph Model for All Classification Tasks, ICLR, 2024.
>
> [3] Huang, et al., PRODIGY: Enabling In-context Learning Over Graphs, NeurIPS, 2023.
>
> [4] Mao, et al., Graph Foundation Models are Already Here, ICML, 2024.
>
> [5] Crammer, et al., Margin analysis of the LVQ algorithm, NeurIPS, 2002.
>
> [6] Sun, et al., All in One: Multi-Task Prompting for Graph Neural Networks, KDD 2023.

---

> ### Author Response · Authors · 2024-08-10
>
> Dear Reviewer U8xo,
>
> Thank you for your comments and suggestions to improve our paper, we would like to check whether our response answers your questions. Following your comments, we comprehensively discuss the relations and differences between computation trees and subgraphs and provide an extensive quantitative study. We look forward to your reply.

---

> > ### Comment · Reviewer_U8xo · 2024-08-13
> > **Official Comment by Reviewer U8xo**
> >
> > Thank you for the detailed response. I raise my score to 6.

---

> ### Author Response · Authors · 2024-08-13
>
> Thanks for your constructive comments and for raising your score! Your review is really supportive of our work!

---

### Official Review · Reviewer_H6gM · 2024-07-17

**Soundness:** 3
**Presentation:** 3
**Contribution:** 3
**Rating:** 7
**Confidence:** 4

**Summary:**

The paper proposes a new graph foundation model based on the tree structure, which is called GFT. GFT leverages computation trees to define tokens within the transferable vocabulary, which improves model generalization and reduce the risk of negative transfers.  Comprehensive experiments and theoretical analyses are provided to demonstrate the effectiveness of GFT across diverse tasks and domains in graph learning. Overall, this is a good work with good motivation, a novel method with theoretical support, comprehensive experiments, and good writing.

**Strengths:**

+ The paper is well motivated by the fact that current graph learning models lack identification of a vocabulary that can encode transferable patterns shared among different tasks and domains. Filling the gap is challenging and meaningful.

+ The proposed GFT model is novel. Unlike existing models, it introduces computation tree as transferable patterns and encodes general knowledge of graph into a tree vocabulary, which is new. It aims to improve the model generalization and reduce the risk of negative transfers.  The theoretical analyses are also provided to support the design of GFT model, which is solid.

+ Extensive experiments over different graph learning tasks (e.g., node classification, link prediction, graph classification) are conducted across many datasets. The model outperforms SOTA baseline methods. As a graph foundation model, it is good to see experiments over cross-domain and cross-task datasets. Many more analytical experiments (including Appendix) are also provided, which is impressive.

+ The paper presentation is good. The organization is clear and easy to follow.

**Weaknesses:**

-  The proposed model assumes tree structure as transferable patterns. Besides the examples (i.e., basic blocks in Fig. 2) shown in the paper, is there any other structures transferable? It is not clear whether these patterns are transferred during the model training or not. How to demonstrate this?
-  As a graph foundation model, besides general and few-shot tasks, it would be better to see experiments on zero-shot task.  Can the proposed model be applied to this scenario and what is performance compared to baseline methods?

**Questions:**

Please see the weaknesses.

**Limitations:**

None.

---

> ### Author Rebuttal · Authors · 2024-08-04
>
> Dear Reviewer H6gM,
>
> We greatly appreciate your acknowledge of our contribution and insightful feedback to help us refine our work. We address your concerns as follows:
>
> > W1: Is there any other structures transferable? How to demonstrate what tokens learn exactly?
> >
>
> Yes, there are other transferable structures. While we did not provide additional visualized examples due to the inherent complexity of graphs, we utilized the Weisfeiler-Lehman (WL) subtree kernel to directly measure the presence of such transferable structures across two graphs. Our experimental results (Figure 3 and Table 1) demonstrate a positive correlation between transferability and the WL subtree kernel value.
>
> Additionally, each token in the vocabulary represents the embedding of certain transferable structures. To elucidate the potential meaning of these tokens, we propose measuring the density of token usage across different datasets. This approach is illustrated in ****Figure 1 of the attached PDF**** in the general response. In this figure, we observe a set of tokens used by all datasets, suggesting they capture general and transferable structures across graphs. Furthermore, some tokens are shared within specific domains or datasets, indicating they maintain domain-specific patterns. It is important to note that our visualization represents an initial step towards understanding the transferable patterns on graphs. Your review has inspired us to further explore this direction in our future work.
>
> > W2: Zero-shot learning setting
> >
>
> We appreciate your suggestion for a comparison of zero-shot learning. Recognizing the importance of zero-shot scenarios for graph foundation models, we have implemented a zero-shot version of our model, GFT. Following current promising approaches [2-4], we use an additional LLM as the task reasoner, utilizing the learned embedding and specific prompts as input. Specifically, we follow the design in [4], using an MLP projector to map the learned embedding and fine-tuning the LLMs with LoRA.
>
> We compare GFT with four baselines that also use LLM reasoners. Models such as GraphGPT [2] and LLAGA [3] focus on single tasks, converting node features and graph substructures for LLM processing. OFA [1] treats subgraphs as transferable patterns and applies graph prompt learning for zero-shot learning, although its performance is limited. Therefore, we implemented an LLM-based OFA (OFA + LLM) similar to our method (GFT + LLM). The results are presented below, with “*” indicating a slightly different experimental setting, such as different pre-training datasets.
>
> |  | Cora | Arxiv | WN18RR |
> | --- | --- | --- | --- |
> | GraphGPT | 18.13* | 64.76* | - |
> | LLAGA | 34.69* | - | - |
> | OFA | 19.31 | 46.19 | 19.98 |
> | OFA + LLM | 30.56 | 64.32 | 27.41 |
> | GFT + LLM | 34.66 | 67.80 | 30.30 |
>
> The advantage of the baseline methods (GraphGPT, LLAGA) is primarily due to their pre-training on a single dataset (or datasets from the same domain), allowing them to learn domain-related patterns for downstream tasks. In contrast, OFA and our GFT are pre-trained on cross-domain and cross-task graphs, acting as general reasoners over graphs. Specifically, the basic OFA’s performance is relatively lower without directly leveraging the power of LLMs in reasoning. However, when using LLMs as reasoners, OFA’s performance significantly improves, though it remains lower than our GFT. This indicates the potential of using computation trees as transferable patterns in learning generalized embeddings.
>
> Best regards,
>
> The authors
>
> ---
>
> References:
>
> [1] Liu, et al., One For All: Towards Training One Graph Model for All Classification Tasks, ICLR, 2024.
>
> [2] Tang, et al., GraphGPT: Graph Instruction Tuning for Large Language Models, SIGIR, 2024.
>
> [3] Chen, et al., LLaGA: Large Language and Graph Assistant, ICML, 2024.
>
> [4] He, et al.,UniGraph: Learning a Cross-Domain Graph Foundation Model From Natural Language, Arxiv, 2024.

---

> > ### Author Response · Authors · 2024-08-11
> >
> > Thanks for your support and constructive comments again! Your review is really helpful for our work!

---

> ### Author Response · Authors · 2024-08-10
>
> Dear Reviewer H6gM,
>
> Thank you for your comments and suggestions to improve our paper, we would like to check whether our response answers your questions. Following your comments, we provide more analysis on transferable structures and conduct experiments on zero-shot settings. We look forward to your reply.

---

> > ### Comment · Reviewer_H6gM · 2024-08-11
> > **Keep my score**
> >
> > I thank the authors' responses, which have addressed my concerns. I would like to keep my acceptance score.

---

### Author Rebuttal · Authors · 2024-08-05

Dear ACs and reviewers,

We thank the reviewers for their feedback and constructive suggestions. The positive feedback has truly inspired us, highlighting the importance of exploring transferable patterns on graphs (H6gM, M9ZQ, hEws), the novelty of our GFT (H6gM, U8xo, M9ZQ), the meaningfulness of cross-domain and cross-task settings (all reviewers), and the soundness of theoretical analysis (all reviewers). In particular, we appreciate Reviewer M9ZQ’s recognition as “the first one demonstrates the potential existence of transferable tokens in text-attributed graphs.”

Here we address some common concerns.

> W1: Differences between computation tree and subgraph

Some reviewers (U8xo, hEws) expressed confusion regarding the differences between computation trees and subgraphs. We provide an illustration in **Figure 2 in the attached PDF** for better clarification. Our concept of computation trees is closely aligned with [1], representing tree-like patterns derived from unfolding the message passing process. Encoding the computation trees of a node is equivalent to encoding the node itself via message passing GNNs, implying that the information in computation trees can be *fully* learned by basic GNNs, demonstrating both learnability and efficiency in encoding computation trees. Moreover, computation trees appear across graph-related tasks. For example, node classification involves classifying the computation tree of a node; edge classification involves classifying the computation tree of a virtual node connecting the end nodes of an edge (u, v); and graph classification involves classifying the computation tree of a virtual node connecting all nodes in the graph. In other words, graph-related tasks can be unified as computation tree-level tasks. Notably, our computation tree can be reinterpreted as a special pattern preserved on the ego-graph of the target node, differing from junction trees or H-trees, which construct additional tree-like graphs to complement the original graph.

Subgraphs, on the other hand, are graph-like substructures within the original graph, such as motifs in molecule graphs. Reference [2] identifies subgraphs as basic patterns across graph-related tasks and reformulates these tasks into subgraph-level tasks. For example, in node classification, they extract the ego-graph around each node and assign the label of the induced graph as the label of the center node, converting node classification into subgraph classification. This process involves (1) extracting ego-graphs around task-relevant nodes and (2) using GNNs to learn graph-level embeddings for classification. However, this extraction process introduces additional time consumption and increased memory usage for storing induced subgraphs. More importantly, the information in subgraphs is not always learnable by basic GNNs, as they cannot detect some critical substructures necessary for learning graph-level embeddings [3,4], reducing the feasibility of using subgraphs to define graph vocabularies.

We provide empirical analysis for better understanding. Efficiency analysis is presented in **Figure 3 of the attached PDF**. Subgraphs generally incur an extra 1/3 time consumption compared to computation trees and encounter out-of-memory errors when batch size exceeds 2048, compared to 8192 for computation trees. The performance comparison is shown in the table below, where the subgraph version (GFT - Subgraph) performs worse than the computation tree version (GFT). We use GAT and GraphMAE as additional baselines and apply linear classifiers on all models for a fair comparison.

||Node|Link|Graph|Avg.|
|---|:---:|:---:|:---:|:---:|
|GAT|74.69|84.55|67.83|75.44|
|GraphMAE|73.98|82.15|62.17|73.07|
|GFT - Subgraph|74.23|86.49|67.89|76.13|
|GFT|75.51|88.99|72.21|78.06|

[1] Chuang, et al., Tree mover’s distance: Bridging graph metrics and stability of graph neural networks, NeurIPS, 2022.

[2] Sun, et al., All in One: Multi-Task Prompting for Graph Neural Networks, KDD 2023.

[3] Chen, et al., Can Graph Neural Networks Count Substructures?, NeurIPS, 2020.

[4] Zhang, et al., Beyond Weisfeiler-Lehman: A Quantitative Framework for GNN Expressiveness, ICLR, 2024

> W2: Zero-shot learning setting

Some reviewers (H6gM and U8xo) suggested a comparison of zero-shot learning. Thus, we follow existing approaches [2-4] by using an additional LLM as the task reasoner to enable zero-shot learning. Specifically, we follow the design in [4] to combine the learned embeddings and specific prompts as input for the LLM and finetune the LLM with LoRA. We select four baselines: GraphGPT [2] and LLAGA [3] designed for a single graph task, OFA designed for cross-tasks, and OFA + LLM leveraging the power of LLM based on OFA, similar to our GFT + LLM. The results are presented below.

||Cora|Arxiv|WN18RR|
|---|:---:|:---:|:---:|
|GraphGPT|18.13|64.76|-|
|LLAGA|34.69|-|-|
|OFA|19.31|46.19|19.98|
|OFA + LLM|30.56|64.32|27.41|
|GFT + LLM|34.66|67.80|30.30|

The advantage of GraphGPT and LLAGA is primarily due to their pre-training on a single dataset (or datasets from the same domain), allowing them to learn domain-related patterns for downstream tasks. In contrast, OFA and our GFT are pre-trained on cross-domain graphs, learning general knowledge rather than specialized knowledge. Specifically, the basic OFA’s performance is relatively lower due to the use of graph prompt learning, which might be less expressive than LLMs. When using LLMs as reasoners, OFA’s performance significantly improves, though it remains lower than our GFT.

[1] Liu, et al., One For All: Towards Training One Graph Model for All Classification Tasks, ICLR, 2024.

[2] Tang, et al., GraphGPT: Graph Instruction Tuning for Large Language Models, SIGIR, 2024.

[3] Chen, et al., LLaGA: Large Language and Graph Assistant, ICML, 2024.

[4] He, et al.,UniGraph: Learning a Cross-Domain Graph Foundation Model From Natural Language, Arxiv, 2024.

Best regards,

The authors

---

### Decision · Program_Chairs · 2024-09-25

**Decision:**

Accept (poster)

**Comment:**

This paper proposes a graph foundation model that uses vector quantization techniques to promote GNN node embeddings (which encode the so-called "computation trees") to be assigned to a limited vocabulary of tree tokens each having a learnable embedding. The tree tokens are shared across different datasets to improve the transferrability of GFM and empirically perform well. Basically this is a good paper for the first time exploring how to enhance the transferrability of graph patterns learned by a GFM by learning a shared vocabulary across graphs.

I have a few suggestions to further improve the readability of the paper:

1. Only until I read all the reviews and rebuttals did I realize many key details of the paper, e.g., the proposed model uses OFA's sentence encoder to align the feature spaces across graph datasets, and that Equation (2) has serious typos, etc. These should be fixed, otherwise careful readers can be frustrated with consistent questions in mind while reading the paper.

2. The statement "Since computation trees are capable of capturing localized patterns [ 51 ], it’s able to represent a graph as a multiset of computation trees without information loss" is incorrect. The WL test exactly represents a graph as a multiset of computation trees while cannot solve the graph isomorphism problem. The encoding of graph into trees is lossy.

3. In response to W4 of U8xo, the statement "However, this limitation arises primarily due to the limited expressiveness of the backbone model (GraphSAGE) rather than our model design... We can address this issue by using more expressive GNN backbones as described in [4]" is incorrect. As pointed out in [1], no matter how expressive the node-level GNN is, encoding nodes independently of each other cannot solve the link-level isomorphism. Therefore, the computation trees have their fundamental limitation in expressiveness. The authors are encouraged to discuss this point as well as other potential improvements in their limitation section (no one exists right now).

[1] Labeling trick: A theory of using graph neural networks for multi-node representation learning. NeurIPS 2021.